# Microenvironment reconstitution of highly active Ni single atoms on oxygen-incorporated Mo$_2$C for water splitting

Mengyun Hou[1,4], Lirong Zheng[2,4], Di Zhao [1] ✉, Xin Tan [3], Wuyi Feng[1], Jiantao Fu[1], Tianxin Wei[1], Minhua Cao [1] ✉, Jiatao Zhang[1] ✉ & Chen Chen [3] ✉

The rational design of efficient bifunctional single-atom electrocatalysts for industrial water splitting and the comprehensive understanding of its complex catalytic mechanisms remain challenging. Here, we report a Ni single atoms supported on oxygen-incorporated Mo$_2$C via Ni-O-Mo bridge bonds, that gives high oxygen evolution reaction (OER) and hydrogen evolution reaction (HER) bifunctional activity. By ex situ synchrotron X-ray absorption spectroscopy and electron microscopy, we found that after HER, the coordination number and bond lengths of Ni-O and Ni-Mo (Ni-O-Mo) were all altered, yet the Ni species still remain atomically dispersed. In contrast, after OER, the atomically dispersed Ni were agglomerated into very small clusters with new Ni-Ni (Ni-O-Ni) bonds appeared. Combining experimental results and DFT calculations, we infer the oxidation degree of Mo$_2$C and the configuration of single-atom Ni are both vital for HER or OER. This study provides both a feasible strategy and model to rational design highly efficient electrocatalysts for water electrolysis.

Electrochemical water splitting is widely regarded as one of the most important and promising pathways to convert renewable energy into high-purity hydrogen and oxygen[1–4]. For a continuous overall water splitting process in practical, pairing the two electrode reactions together in an integrated electrolyser is difficult because of the mismatch of pH ranges in which these catalysts are stable and remain the most active[5–8]. In addition, producing different catalysts for both monofunctional catalysts requires different equipment and processes, which could increase the cost compared to bifunctional catalysts. Therefore, compared to monofunctional catalysts, bifunctional catalysts are relatively favorable options for overall water decomposition[9]. Currently, Pt-based materials and Ir/Ru-based materials are the benchmark in HER and OER catalysts, respectively. Nevertheless, their scarcity and high price hinder their development and wide applications[10–13]. Meanwhile, only a few OER catalysts (e.g. IrO$_2$ and RuO$_2$) are stable in acidic solutions, thus designing cost-effective and

earth-abundant bifunctional electrocatalysts with high efficiency of overall water splitting in the same alkaline conditions are of great significance and urgency[7,14,15].

Single-atom catalysts (SACs) featuring maximizing the atomic utilization rates and the synergistic effects between single-atom sites and its support are considered as the most promising strategy for designing cost-effective and earth-abundant bifunctional electrocatalysts[16,17]. Currently, most SACs are based on carbonaceous substrates, such as amorphous carbon, graphene, with isolated metal atoms coordinated by intrinsic defects or heteroatoms. However, such catalysts show insufficient activity when applied in HER, especially in OER, while they exhibited obvious improvement in electrocatalytic O$_2$/CO$_2$ reduction reactions. So, it is a critical challenge to develop SACs with new highly conductive support. Transition metal-based carbides, such as molybdenum carbide (Mo$_2$C), had metal-like electric conductivity and received special attention as potential bifunctional

[1]Key Laboratory of Cluster Science, Ministry of Education of China, Beijing Key Laboratory of Photoelectronic/Electrophotonic Conversion Materials, School of Chemistry and Chemical Engineering, Beijing Institute of Technology, Beijing 100081, China. [2]Beijing Synchrotron Radiation Facility, Institute of High Energy Physics, Chinese Academy of Sciences, Beijing 100049, China. [3]Engineering Research Center of Advanced Rare Earth Materials, Department of Chemistry, Tsinghua University, Beijing 100084, China. [4]These authors contributed equally: Mengyun Hou, Lirong Zheng. ✉e-mail: dizhao@bit.edu.cn; caomh@bit.edu.cn; zhangjt@bit.edu.cn; cchen@mail.tsinghua.edu.cn

electrocatalysts for water splitting. However, the water-splitting catalytic performance of pure $Mo_2C$ is far from satisfactory due to the weak adsorption strength of oxygen species and the slow dissociation kinetics of $H_2$ and other reaction intermediates[18,19]. To address this challenge, one of the most effective strategies is to decorate with transition metals (Ni, Co, Pt, and W)[20–22]. In addition, the bridging bond $M_1$-O-$M_2$ (M=metal) between the substrate material and single atoms can produce strong electronic coupling via the surface, resulting in effective charge transfer, which can optimize adsorption/desorption behavior of the active site for oxygen intermediates, hydrogen intermediates[23]. For example, Suryanto group reported that the surface unique $M_1$-O-$M_2$ configuration could modify Gibbs free energy absorption of reaction intermediates (O* and OH*), then breaking the scale relationship and promoting OER and HER activities[24–26]. So, it is feasible and challenging to construct atomically distributed metal centers on surface-oxygen-abundant $Mo_2C$ to effectively modulate the adsorption/desorption behavior of the key intermediates during the water splitting process.

Furthermore, with insights into the electrocatalytic process, studies have evidenced that many electrocatalysts undergoes reconstitution during the so-called activation process or the initial cyclic voltammetry measurement. As for the HER example, Li et al. demonstrated that the $V_O$-Ru/HfO$_2$-OP occurred structural change during the HER activation process, resulting in high catalytic activity as well as stability[27]. In addition, the Ni@Ni/NiO$_x$ actual active sites for HER were generated by core@shell structure of Ni@O-Ni pre-electrocatalyst through the potential-driven reconstitution[28]. Besides, Wang group had demonstrate that the active HER species of Co@CoFe-P NBs arose from in situ reconstructed P-Co-O-Fe-P configurations with low-valence metal sites ($M^0/M^+$), which can decrease the energy barriers for water dissociation and adsorption of H* intermediates[29]. In sharp contrast, their OER activity arose from the oxygen-bridged, high-valence $Co^{IV}$-O-$Fe^{IV}$ moieties. Due to the redox process in OER, the catalyst surface reconstitution is more likely to accompanied. Most of the catalysts are fully oxidized into transition metal oxide or hydroxides, or their surfaces can be reconstructed into different coordination environments. Zhou et al. identified that the CoNi$_{0.5}$P catalyst was transformed into the low-crystalline CoNi$_{0.25}$O$_x$(OH)$_y$ during the OER process and found that the CoNi$_{0.25}$O$_x$(OH)$_y$ serve as the reactive species for OER rather than the original CoNi$_{0.5}$P species[30]. Meanwhile, Li group reported the reconstitution of WC$_x$-FeNi catalysts, that is, O-bridged FeNi moieties were formed after electrochemical treatment, which provided the high oxygen-evolving catalytic activity[31]. All the pioneering works mentioned above have demonstrated, that the ability of the catalyst to "evolve" during the reaction may be a key feature of highly active OER catalysts. So, understanding of the structural transformation of the catalysts is crucial to better unveil the real catalytic mechanism and design novel electrocatalysts, although they have rarely been studies for SACs[32].

Here, we primarily motivated to use $Mo_2C$ with the partially oxidized surface as a support material to stabilize Ni single atoms. The resulting materials possess a thin layer of atomic Ni sites, which interacted with the oxidized surface of $Mo_2C$ via Ni-O-Mo bridge bonds (Ni$_{SA}$-O/$Mo_2C$) and exhibited good catalytic activity for HER, OER, and overall water splitting. By ex situ synchrotron X-ray absorption spectroscopy and electron microscopy observations, we found, the configuration of atomically dispersed Ni in Ni$_{SA}$-O/$Mo_2C$ electrocatalyst underwent transformation and O/$Mo_2C$ support were overoxidized or dissolved to varying degrees. After HER, the coordination number of Ni-O and the bond length of Ni-Mo (Ni-O-Mo) were all increased, but Ni species still remained atomically dispersed form. However, after OER, the coordination number of Ni-O increased, the coordination number of Ni-Mo (Ni-O-Mo) decreased but new Ni-Ni (Ni-O-Ni) bonds appeared, indicating the atomically dispersed Ni were agglomerated into very small clusters

under applied potentials. Combining the experimental results and DFT calculations, we found partially oxidized $Mo_2C$ will be beneficial to HER performance. However, compared with the original Ni$_{SA}$-O/$Mo_2C$, the potential-driven reconstituted small Ni clusters with optimal Ni-O, Ni-Mo and Ni-Ni coordination structures are more conducive to catalyzing the production of $O_2$.

## Results

### Material synthesis and characterization

The Ni$_{SA}$-O/$Mo_2C$ nanospheres were synthesized by the following steps: (1) preparation of Mo-based organic nanosphere precursors by a facile hydrothermal method; (2) adsorption of the Ni metal ions via impregnation method; (3) heat treatment at certain temperatures under air and $H_2$/Ar atmospheres. During the pyrolysis process, the Mo-based organic precursors were simultaneously converted to $Mo_2C$ nanocrystals with Ni ions fixed on their surfaces, which embedded in a uniform carbon substrate. Transmission electron microscopy (TEM) images of Fig. S1 shows that the amorphous Mo-based precursor is a nanosphere structure with rough surfaces with a diameter of about 200 nm, and the morphology does not change significantly after the adsorption of Ni ions on the surface (Fig. S2). After pyrolysis, an obvious double-shelled hollow structure can be observed (Fig. 1a, b). The mechanism of double-shelled hollow structure formation has been studied in detail (Figs. S3–S6) and its possible schematic illustration is shown in Fig. S7, which is mainly based on the early oxidation of air and non-equilibrium non-uniform shrinkage caused by heat treatment[33,34]. During the formation of double-shelled hollow structure, the conditions of air and $H_2$/Ar atmospheres are both vitally important. In addition, the specific surface areas obtained from $N_2$ adsorption-desorption isotherms show a significant increase from 13.9 for O/$Mo_2C$ to 64.3 m$^2$ g$^{-1}$ for Ni$_{SA}$-O/$Mo_2C$ (Fig. S8), which is probably due to the volatilization of nitrate from Ni sources. X-ray diffraction (XRD) patterns (Fig. S9) indicate that the diffraction of Ni$_{SA}$-O/$Mo_2C$ can be well indexed to pure $Mo_2C$ phase (JCPDS card no.35-0787)[35]. No peaks of Ni metals, metal oxide can be observed. The selected area electron diffraction (SAED) pattern (insert in Fig. 1b) indicates the polymorphism of the Ni$_{SA}$-O/$Mo_2C$.

Energy-dispersive X-ray spectroscopy (EDS) mapping images reveal that Mo, Ni, and O elements almost have the same double-shelled hollow shape and C elements is uniformly distributed in an individual Ni$_{SA}$-O/$Mo_2C$ nanospheres (insert in Fig. 1c, and Fig. 1d). Combined with the HR-TEM magnified image in Fig. S10, we further confirm that $Mo_2C$ nanocrystals with Ni ions were embedded in a uniform carbon substrate. To further identify the exact location of single Ni atoms in the $Mo_2C$ nanospheres, we performed sub-angstrom-resolution high-angle annular dark field aberration-corrected (HAADF)-scanning TEM (STEM) (Fig. 1e, f). Figure 1e clearly detected the lattice fringes with an interplanar distance of 0.228 nm and 0.135 nm, corresponding to the (101) and (103) plane of a hexagonal β-$Mo_2C$, respectively. Detailed observation of the edge of the bright Mo atom array reveals an amorphous or low crystallinity molybdenum oxide (MoO$_x$) layer (around the yellow line in Fig. 1f) due to surface oxidation[36,37]. In addition, due to the different Z-contrast, the darker spots (red circles in Fig. 1f) on the edge of brighter spots (Mo atoms) can be identified as isolated Ni single atoms[38]. The intensity profiles at the location of position g1 and g2 marked by yellow circles in Fig. 1f including darker spots and brighter spots of surface MoO$_x$ layer further confirm the presence of isolated Ni$_{SA}$ single atoms (Fig. g1 and g2)[8]. For comparison, the control electrocatalysts of Ni$_{1.5}$-O/$Mo_2C$ and Ni$_{4.5}$-O/$Mo_2C$ were prepared with varying amounts of Ni. XRD patterns and TEM images display the same crystal structure and morphologies as those of Ni$_{SA}$-O/$Mo_2C$ (Figs. S9, S11). Their Ni-loading contents detected by inductively coupled plasma atomic emission spectroscopy (ICP-AES) are 0.75 wt%, 0.98 wt% and 1.33 wt%, respectively (Table S1).

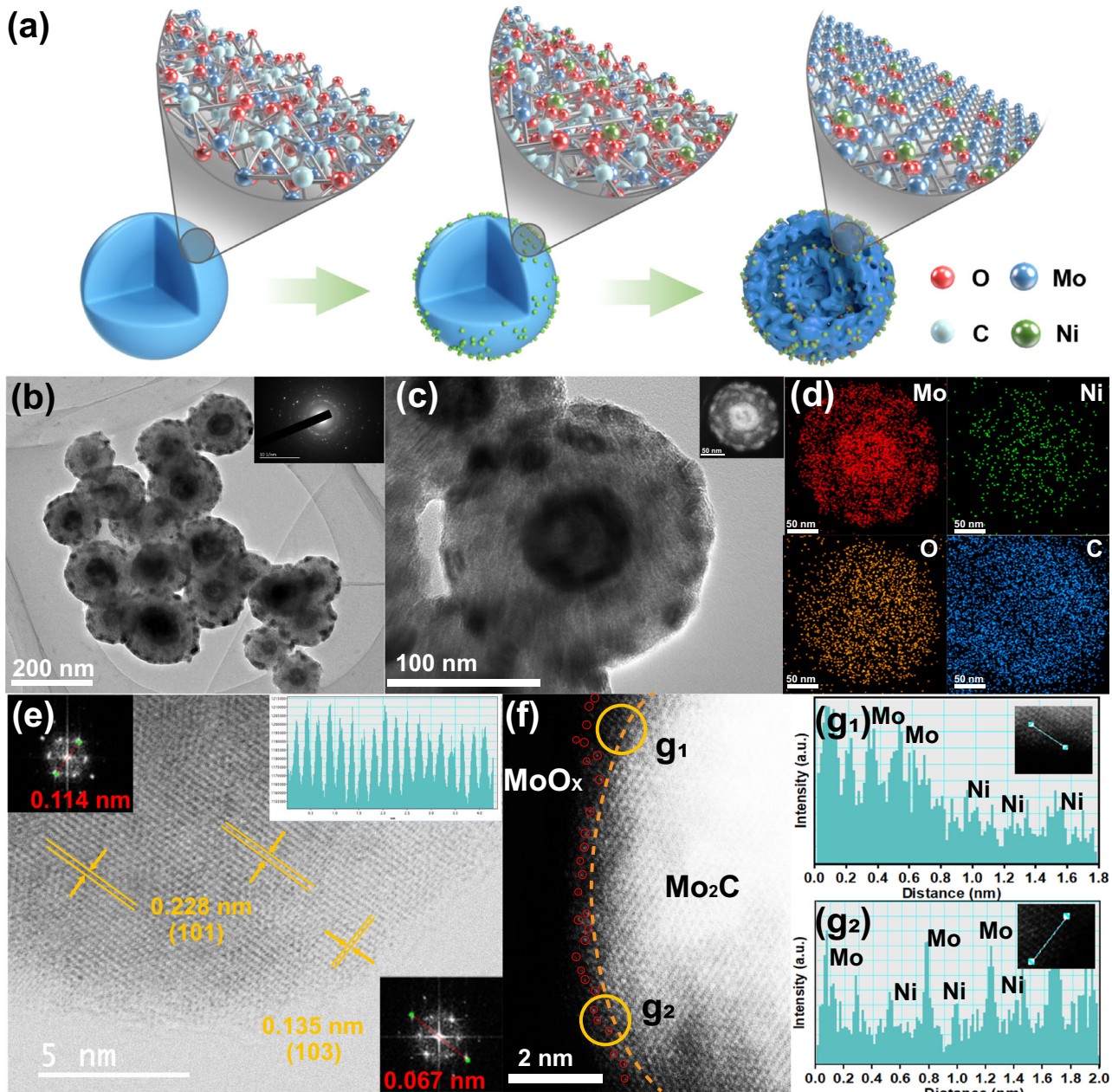

**Fig. 1 | Morphology and structure of the materials. a** Schematic illustration of the fabrication process of the $Ni_{SA}$-O/$Mo_2C$ electrocatalyst. **b, c** HR-TEM image of $Ni_{SA}$-O/$Mo_2C$. Insets in (**b**): SAED pattern. **d** Elemental mapping of Mo, Ni, O, C for an individual $Ni_{SA}$-O/$Mo_2C$ nanospheres (inset in Fig. 1c), respectively. **e** HAADF-STEM image and **f** corresponding enlarged view of the $Ni_{SA}$-O/$Mo_2C$. The bright dots highlighted with red in (**f**) marks the Ni single-atoms in $Ni_{SA}$-O/$Mo_2C$. **g** Lineal contrast analysis for selected area of g1 and g2 at the edge of Fig. 1f.

Since the coordination environment determines the electronic structure, the valence states of Mo, and Ni have been studied by X-ray photoelectron spectroscopy (XPS) (Figs. S12-13). As shown in Fig. 2a, the high-resolution Mo 3d spectra of pure $Mo_2C$ were deconvoluted into six peaks, corresponding to $Mo^{2+}$ (228.6 eV, 231.7 eV), $Mo^{4+}$ (229.0 eV, 232.0 eV) and $Mo^{6+}$ (232.8 eV, 235.8 eV) species[39]. $Mo^{4+}$ and $Mo^{6+}$ can be assigned to molybdenum oxides due to surface oxidation of Mo species, indicating a thin surface layer of $MoO_x$ were formed outside $Mo_2C$ electrocatalyst (denoted as O/$Mo_2C$)[37]. This is well consistent with HAADF-STEM analysis (Fig. 1e-f). It is worth mentioning that, the surface oxide layer can be controlled by the atmosphere and temperature to which the $Ni_{SA}$-O/$Mo_2C$ is exposed. For example, when the newly reduced $Ni_{SA}$-O/$Mo_2C$ were taken out of the tube furnace at 60 °C in air, its surface would be much more oxidized compared with the main sample of $Ni_{SA}$-O/$Mo_2C$. While, when the newly reduced $Ni_{SA}$-O/

$Mo_2C$ were protected in Ar atmosphere for 12 h before taking out the furnace, its surface would be much lower oxidized compared with the main sample of $Ni_{SA}$-O/$Mo_2C$. (Fig. S13a, b). In addition, after etching the surface oxide layer, the main phase of $Mo_2C$ gradually leaks out, which is demonstrated by XPS results with different etching depths (Fig. 2a, Fig. S13c). The XPS spectrum of C 1s (Fig. S14a) is fitted into four different signals at 283.9, 284.8, 285.8, and 288.9 eV, which are attributed to C-Mo, C−C/C=C, C−O, and O−C=O, respectively[37]. After loading of Ni atoms, the binding energy of Mo 3d XPS spectra for $Ni_{SA}$-O/$Mo_2C$ display a positive shift about 0.12 eV compared with that of pure O/$Mo_2C$, indicating the charge migration caused by the atomic Ni incorporation[13]. This is further proved by the positive shift of the main peak of O XPS for $Ni_{SA}$-O/$Mo_2C$ compared with that of pure O/$Mo_2C$ (Fig. S14b). The Ni 2p XPS spectrum can be split into six peaks, of which the peaks at 856.4 and 874.2 eV are assigned to $Ni^{2+}$ caused by the

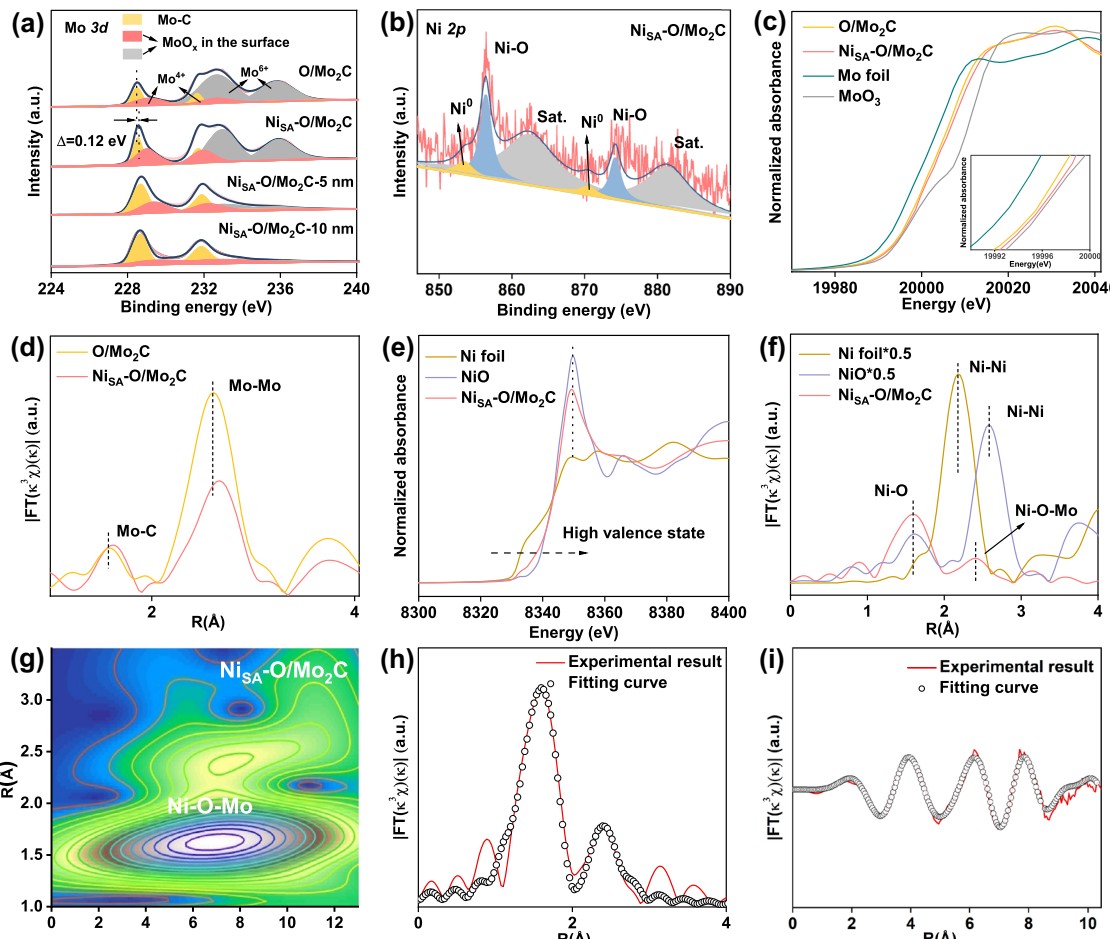

**Fig. 2 | Structural characterization. a** XPS spectra of Mo *3d*. **b** XPS spectra of Ni *2p*. **c** Mo K-edge XANES spectra. Inset in (**c**) is the partial enlargement of the pre-edge peak. **d** The FT-EXAFS curves for the as-prepared samples. **e** Ni K-edge spectra. **f** The corresponding FT-EXAFS curve. **g** The wavelet transform (WT) of the Ni$_{SA}$-O/Mo$_2$C. **h, i** Ni K-edge EXAFS of Ni$_{SA}$-O/Mo$_2$C in R and k spaces.

bonding between Ni and adsorbed oxygen, and the peaks at 862.4 and 881.4 eV correspond to the satellite peaks of Ni (Fig. 2b). Only a small amount of Ni$^0$ can be observed at 853.5 and 870.5 eV, which indicates that the Ni atoms are mainly coordinated with the surrounding oxygen atoms[8]. It should be noted that, due to the Ni content in the Ni$_{1.5}$-O/Mo$_2$C is lower than Ni$_{SA}$-O/Mo$_2$C, the peak of Ni$^0$ is not obvious, while Ni$^{2+}$ is observed at 862.4 and 881.4 eV, indicating that Ni in the Ni$_{1.5}$-O/Mo$_2$C sample is almost entirely bonded to the surrounding oxygen atoms. While, the proportion of Ni$^0$ peak in Ni$_{4.5}$-O/Mo$_2$C is significantly larger than that in Ni$_{SA}$-O/Mo$_2$C (Fig. S15), which is probably due to the excessive Ni aggregates into Ni clusters or nanoparticles.

To assess the electronic structure and local environment of the Ni and Mo in Ni$_{SA}$-O/Mo$_2$C, X-ray absorption near-edge spectroscopy (XANES) and extended X-ray absorption fine structure (EXAFS) were investigated. The Mo K-edge XANES spectra are shown in Fig. 2c. Compared with pure O/Mo$_2$C, the pre-edge of Ni$_{SA}$-O/Mo$_2$C was higher, suggesting that the average oxidation state of Mo in the Ni$_{SA}$-O/Mo$_2$C was more positive, which is consistent with the aforementioned XPS results. The Fourier transform (FT) extended EXAFS (Fig. 2d) of the pure O/Mo$_2$C and Ni$_{SA}$-O/Mo$_2$C possessed two peaks. The strongest peak is located at ~2.6 Å, corresponding to the Mo-Mo bonds. The other peak is located at ~1.6 Å, which is associated with the Mo-C bonds[35]. Compared with pure O/Mo$_2$C, the introduction of Ni would extend the Mo-Mo and Mo-C bonding length due to the interaction between Ni and Mo$_2$C. The Ni K-edge pre-edge of Ni$_{SA}$-O/Mo$_2$C is located between the Ni and NiO (Fig. 2e), which indicates that the valence of Ni in Ni$_{SA}$-O/Mo$_2$C is

situated between 0 and +2. The Ni K-edge XANES spectra and the corresponding k-space spectra of Ni$_{SA}$-O/Mo$_2$C closely resemble that of NiO reference (Fig. S16), suggesting the formation of surface Ni-O[36]. The Fourier transformed (FT) k$^3$-weighted EXAFS spectrum in R space further confirmed the formation of Ni-O on the amorphous thin layer MoO$_x$ on the surface of Mo$_2$C. As shown in Fig. 2f, one notable peak at ~1.58 Å contributed by the Ni-O peak is observed. The FT k$^3\chi$(k) of Ni foil shows a peak at ~2.18 Å and NiO displays a peak at ~2.58 Å, both corresponding to the Ni-Ni interaction. Ni$_{SA}$-O/Mo$_2$C does not have these peaks, which demonstrates that Ni atoms should exist in the single-atomic form on the amorphous thin layer MoO$_x$. In addition, there is a scattering peak at around 2.4 Å, which could be due to the bonding of Ni−Mo. Combined with aforementioned results of the oxidized surface of Ni$_{SA}$-O/Mo$_2$C, Ni−O−Mo bonding should dominate on the surface.

We have also performed wavelet transform (WT) analysis of Ni$_{SA}$-O/Mo$_2$C for verifying the coordination environments of Ni-O-Mo bonding, which allows information to be displayed in both R-space and K-space. As shown in Fig. 2g, a strong WT signal focused at 6.91 Å$^{-1}$, which was derived from Ni-O-Mo contribution. To acquire the precise coordination configuration of Ni atoms in Ni$_{SA}$-O/Mo$_2$C, the quantitative EXAFS fitting is performed to extract the structural parameters and the results are presented in Fig. 2h, i and summarized in Table S3. The best-fitted result reveals Ni atom is coordinated with O with a coordination number of 3.9 at 2.01 Å and as well coordinated with Mo with a coordination number of 5.0 at 2.98 Å in the second or higher shell. In addition, the Ni$_{4.5}$-O/Mo$_2$C with higher Ni content has a weak

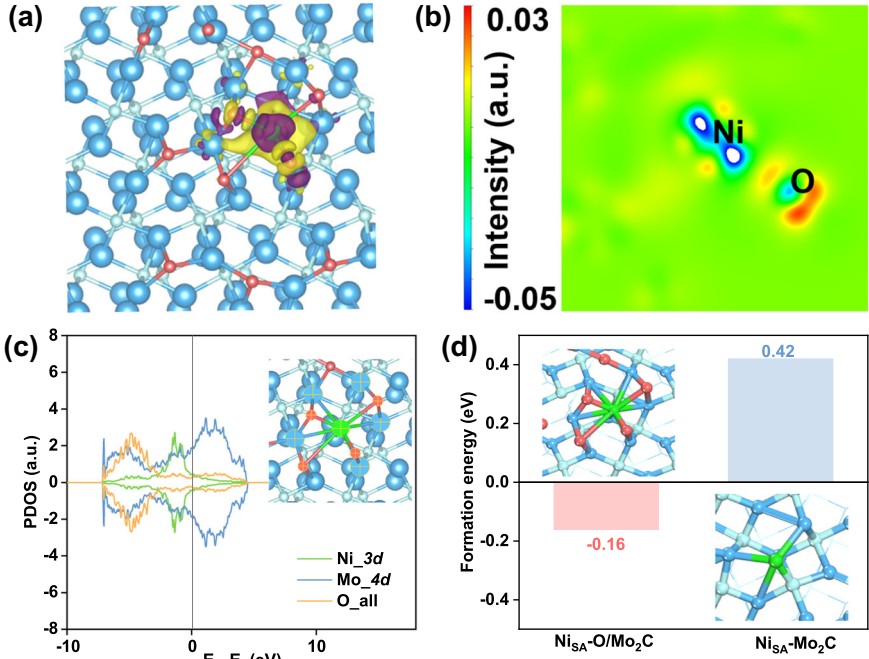

**Fig. 3 | DFT calculation.** The two plots of charge density differences of $Ni_{SA}$-O/ $Mo_2C$: top view of 3D plot (**a**) and 2D display (**b**). The isosurface level set to 0.003 eÅ$^{-3}$, where charge depletion and accumulation were depicted by purple and yellow, respectively. **c** PDOS graph of *3d* orbitals of Ni, 4d orbitals of Mo, and all orbitals of O atoms specified by cross marks in $Ni_{SA}$-O/$Mo_2C$. **d** The chemisorption energies of Ni atom on surfaces of $Ni_{SA}$-O/$Mo_2C$ and $Ni_{SA}$-$Mo_2C$.

peak at 2.18 Å corresponding Ni-Ni signal (Fig. S17), which is well consistent with the results of XPS.

In order to further study the mechanism of Ni atoms binding on the oxygen-incorporated $Mo_2C$, we did a series of DFT calculations. According to the previous results of structural analysis, we chose plane (101) of $Mo_2C$ as the reaction surface. Then, calculations were made on the absorption of oxygen atoms on the surface, and we determined the most stable structure (Fig. S18) by comparing the values of their total energy. So, we attached some oxygen atoms to the facet by the most solid adsorption to simulate a partially oxidized surface of O/$Mo_2C$. According to the above data, we adsorbed Ni atom bonding with four oxygen atoms and one carbon atom at the facet, respectively, to obtain calculation models $Ni_{SA}$-O/$Mo_2C$ and $Ni_{SA}$-$Mo_2C$ (Fig. S19). Firstly, we performed calculations of the charge density difference between Ni and facet, as shown in Fig. 3a, b. It could be clearly seen that obvious tendency to transfer electrons from Ni site to surrounding atoms, indicating the interaction between Ni with facet. This is also confirmed by Fig. 3c, because of overlaps between the orbitals of Ni, O, and Mo atoms (marked by cross) around the Fermi level. So, we concluded that Ni atoms could be stably adsorbed on the partially oxidized facet of $Mo_2C$. In addition, the value of adsorption energy at $Ni_{SA}$-O/$Mo_2C$ (−0.16 eV) was significantly lower than $Ni_{SA}$-$Mo_2C$ (0.42 eV) (Fig. 3d), further indicating that the bonding of Ni-O-Mo was more stable.

## Electrocatalytic performance

To examine the electrochemical OER catalytic activities in 1 M KOH using a three-electrode set-up. As shown in Fig. 4a, the $Ni_{SA}$-O/$Mo_2C$ exhibits an overpotential (η) of 299 mV at 10 mA cm$^{-2}$, which significantly exceeds those of $Ni_{1.5}$-O/$Mo_2C$ (379 mV), $Ni_{4.5}$-O/$Mo_2C$ (337 mV), pure O/$Mo_2C$ (547 mV) and commercial $IrO_2$ catalyst (386 mV). From these η values, we can see that the higher loadings of Ni results in better performance due to the higher density of catalytic reactive centers. However, the catalysts cannot perform even better when the Ni loading is excessive. In addition, the overpotential of $Ni_{SA}$-O/$Mo_2C$ catalyst requires only 430 mV at the current density of 200 mV cm$^{-2}$, even far less than 670 mV required by commercial $IrO_2$ catalyst,

demonstrating a huge improvement of the OER activity under the high current density. In addition, Fig. 4a shows the Ni oxidation peak, corresponding to the oxidation Ni from low valence states ($Ni^0$) to high valence states ($Ni^{2+}$/$Ni^{3+}$)[24]. Tafel slopes were calculated to unveil that $Ni_{SA}$-O/$Mo_2C$ endows a smaller Tafel slope (89.36 mV dec$^{-1}$), much lower than those of $Ni_{1.5}$-O/$Mo_2C$ (99.89 mV dec$^{-1}$), $Ni_{4.5}$-O/$Mo_2C$ (95.22 mV dec$^{-1}$), pure O/$Mo_2C$ (176.28 mV dec$^{-1}$) and even commercial $IrO_2$ (155.77 mV dec$^{-1}$), which is a sign of its fast electron transfer and superior mass transport properties (Fig. 4b)[31]. Motivated by the promising OER performance of $Ni_{SA}$-O/$Mo_2C$, the electrocatalytic HER was also investigated in the same 1.0 M KOH environment. Figure 4c shows the Pt/C catalyst exhibits the best HER activity, whereas $Ni_{SA}$-O/$Mo_2C$ shows competitive performance with an overpotential of 133 mV at a current density of 10 mA cm$^{-2}$, which is lower than those of $Ni_{1.5}$-O/ $Mo_2C$ (163 mV), $Ni_{4.5}$-O/$Mo_2C$ (143 mV) and pure O/$Mo_2C$ (182 mV). Furthermore, the Tafel slope of the $Ni_{SA}$-O/$Mo_2C$ is 83.6 mV dec$^{-1}$, less than those of $Ni_{1.5}$-O/$Mo_2C$ (93.64 mV dec$^{-1}$), $Ni_{4.5}$-O/$Mo_2C$ (92.36 mV dec$^{-1}$) and pure O/$Mo_2C$ (121.59 mV dec$^{-1}$) (Fig. 4d). It is worth noting that in order to exclude the role of Ni atoms on carbon substrate, we set up two control samples: Ni atoms on carbon substrate obtained via etching $Mo_2C$ particles in $Ni_{SA}$-O/$Mo_2C$ ($Ni_{SA}$/$C_{etch}$) and Ni atoms on pure carbon spheres (Ni/C) synthesized by the same method with $Ni_{SA}$-O/$Mo_2C$. As can be seen from XRD patterns, TEM, and element mappings (Fig. S20), both samples only show carbon peaks and uniform Ni element distribution. Electrochemical tests showed that compared with $Ni_{SA}$-O/$Mo_2C$, their HER and OER catalytic activity can be ignored (Fig. S21), indicating the catalytic activity of $Ni_{SA}$-O/$Mo_2C$ comes from the Ni single atoms anchored on oxygen-incorporated $Mo_2C$. Impressively, $Ni_{SA}$-O/$Mo_2C$ outperformed most of the non-precious metal SACs in alkaline media (Fig. 4e, Tables S4-5), proving $Ni_{SA}$-O/ $Mo_2C$ is a potentially highly efficient bifunctional catalyst. To investigate the electrode kinetics under the catalyst process, electrochemical impedance spectroscopy (EIS) measurements were carried out. Fig. S22 displays a favorable charge transfer resistance ($R_{ct}$) of $Ni_{SA}$-O/ $Mo_2C$ ($R_{ct}$=12 Ω) as compared with other control samples[40]. Besides, the electrochemical active surface area (ECSA) of all samples was

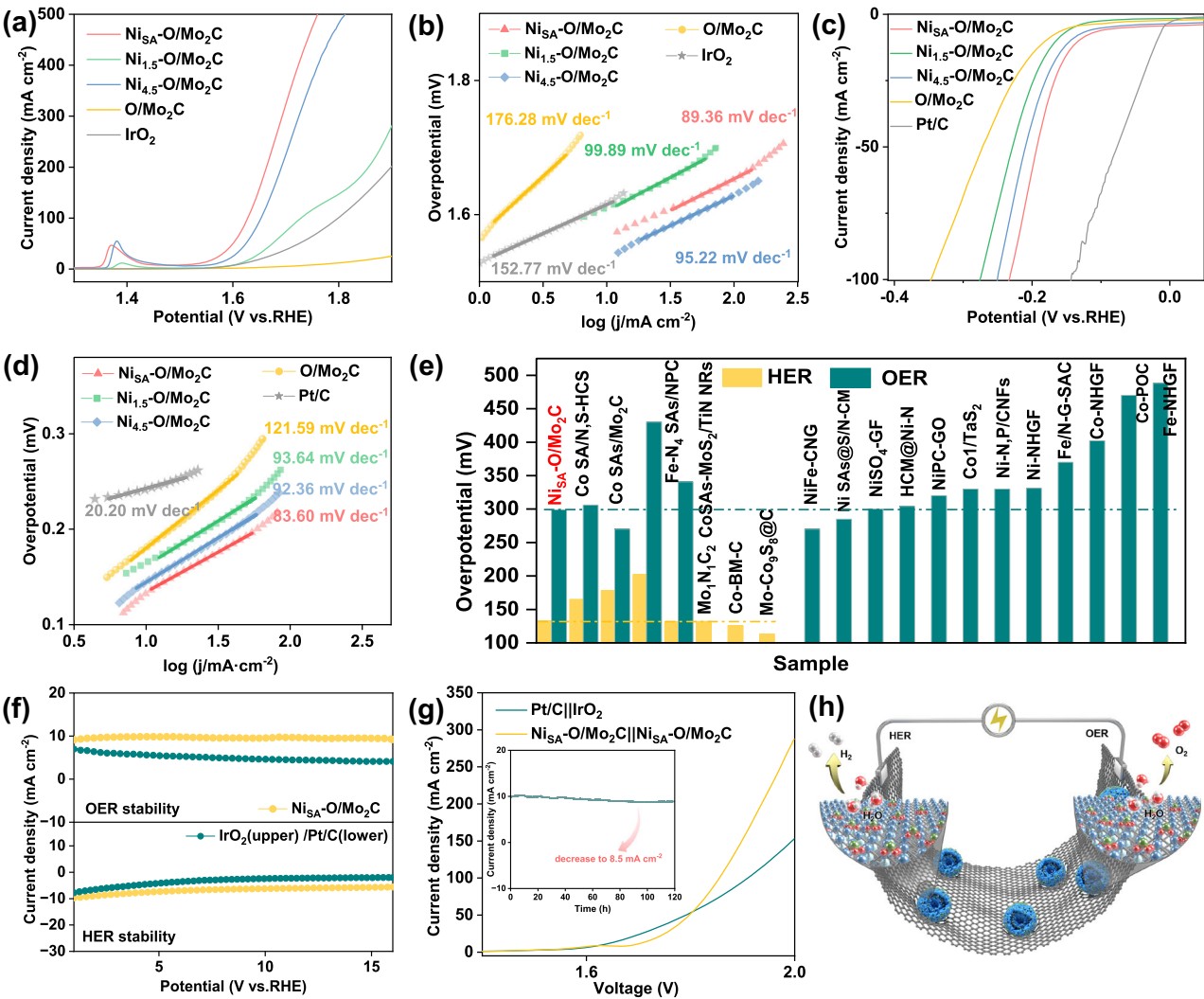

**Fig. 4 | HER, OER and overall water splitting performance of the $Ni_{SA}$-O/$Mo_2C$ catalyst in 1.0 M KOH (pH=14) with iR compensation. a** Polarization curves and **b** Tafel plots for $Ni_{SA}$-O/$Mo_2C$, $Ni_{1.5}$-O/$Mo_2C$, $Ni_{4.5}$-O/$Mo_2C$ and pure O/$Mo_2C$, commercial $IrO_2$ for OER (loading amount 0.74 mg cm$^{-2}$). **c** Polarization curve and **d** Tafel plots for $Ni_{SA}$-O/$Mo_2C$, $Ni_{1.5}$-O/$Mo_2C$, $Ni_{4.5}$-O/$Mo_2C$ and pure O/$Mo_2C$, 20% Pt/C for HER (loading amount 0.7 mg cm$^{-2}$). **e** Comparison of the overpotentials of $Ni_{SA}$-O/$Mo_2C$ and recently reported non-precious metal SACs at 10 mA cm$^{-2}$ in 1.0 M KOH. The histograms in yellow and green denote the HER, OER performance, respectively. **f** Current density versus time (**i**–**t**) curves for HER and OER. **g** Polarization curves of $Ni_{SA}$-O/$Mo_2C$||$Ni_{SA}$-O/$Mo_2C$ and Pt/C||$IrO_2$ couples for overall water splitting in 1.0 M KOH (loading amount 0.74 mg cm$^{-2}$). Inset **g** is the long-term durability of overall water splitting at a current density of 10 mA cm$^{-2}$. **h** Illustration of the $Ni_{SA}$-O/$Mo_2C$ electrodes for overall water splitting.

reflected from the double layer capacitance ($C_{dl}$) (Fig. S23). $Ni_{SA}$-O/$Mo_2C$ catalyst has the highest $C_{dl}$ value, which indicates that $Ni_{SA}$-O/$Mo_2C$ has more exposed active surface area than other samples. Since the specific activity reflects the intrinsic activity of the catalysis, ECSA-normalized LSV curves for both HER and OER were shown in Fig. S24. The specific activity (SA) normalized by ECSA shows that $Ni_{SA}$-O/$Mo_2C$ exhibits the best electrocatalytic activity toward both OER and HER compared to other catalysts following the order of $Ni_{SA}$-O/$Mo_2C$ >$Ni_{4.5}$-O/$Mo_2C$ >$Ni_{1.5}$-O/$Mo_2C$>O/$Mo_2C$. In particular for OER, at the overpotential of 370 mV (Fig. S24d), $Ni_{SA}$-O/$Mo_2C$ exhibits the SA of 8.71 mA cm$^{-2}_{ECSA}$, which is higher than those of $Ni_{4.5}$-O/$Mo_2C$ (4.67 mA cm$^{-2}_{ECSA}$), $Ni_{1.5}$-O/$Mo_2C$ (2.26 mA cm$^{-2}_{ECSA}$) and O/$Mo_2C$ (0.52 mA cm$^{-2}_{ECSA}$). These results indicate incorporation of Ni not only improves the density of active sites but also intrinsically improves HER and especially OER activity of $Ni_{SA}$-O/$Mo_2C$ catalyst. The durability of the catalyst is an important indicator, especially for SACs. For OER, long-term electrolysis at a constant current has been performed under a constant current of 10 mA cm$^{-2}$ by chronoamperometry for 16 h (Fig. 4f), which proved that the currents of $Ni_{SA}$-O/$Mo_2C$ first increase and then maintain stability for a long time. In contrast, after 16 h, the

commercial $IrO_2$ catalyst shows distinctly difference in the required overpotential. For HER, although it was much more stable than 20 wt % Pt/C, the current density of the $Ni_{SA}$-O/$Mo_2C$ catalyst decreased slightly after 16 h.

Furthermore, we assembled an electrolyzer with 1.0 M KOH electrolyte using $Ni_{SA}$-O/$Mo_2C$ as bifunctional electrodes ($Ni_{SA}$-O/$Mo_2C$||$Ni_{SA}$-O/$Mo_2C$). Pt/C||$IrO_2$ was also tested for comparison. Figure 4g gives a comparison of LSV curves of $Ni_{SA}$-O/$Mo_2C$||$Ni_{SA}$-O/$Mo_2C$ and Pt/C||$IrO_2$. The cell voltage required to reach 10, 100, and 150 mA cm$^{-2}$ for $Ni_{SA}$-O/$Mo_2C$ are 1.69 V, 1.86 V, and 1.90 V, while Pt/C||$IrO_2$ requires 1.64 V, 1.91V and 1.93V, respectively. The result shows that the electrocatalytic overall splitting performance of $Ni_{SA}$-O/$Mo_2C$||$Ni_{SA}$-O/$Mo_2C$ is comparable to that of Pt/C||$IrO_2$ especially under the high current density and superior to most reported single-atom catalysts (Table S6). In the stability test, the current density has only decreased from 10 mA cm$^{-2}$ to 8.5 mA cm$^{-2}$ after 120 h, indicating its good stability as a bifunctional electrocatalyst (inset in Fig. 4g). Moreover, the obvious bubbles were observed at the cathode and anode in the long-time stability testing, corresponding to $H_2$ and $O_2$, respectively (Fig. 4h and Fig. S25).

## Potential-driven new active interfaces generation

In order to reveal the different stability phenomena of HER and OER, a series of characterizations were carried out to probe the identity of the real catalytically active species. The XRD pattern (Fig. S26a) indicates that the surface of $Ni_{SA}$-O/$Mo_2C$ undergoes structural reorganization to form different phases after the different electrochemical reactions. After electrochemical HER, the $Ni_{SA}$-O/$Mo_2C$ catalyst maintains the diffraction peaks of $Mo_2C$, but other weaker peaks of $MoO_3$ (JCPDS card no.35-0609) and $MoO_4^{2-}$ (JCPDS card no.29-1021) appear. In contrast, for OER, all the XRD peaks could be assigned to the crystal structures of $MoO_3$ (JCPDS card no. 05-0508 and 35-0609) and $MoO_4^{2-}$ without $Mo_2C$ peaks, indicating the $Mo_2C$ were completely disintegrated. Fig. S27 shows the Mo $3d$ spectra of $Ni_{SA}$-O/$Mo_2C$ after reactions, the major Mo peaks can be assigned to $Mo^{6+}$ without low valence state of Mo, suggesting the average oxidation state of Mo are both higher than that of $Ni_{SA}$-O/$Mo_2C$, that is to say the surface of the catalysts both have obvious oxidation behavior[41,42]. However, after HER reaction, the $Mo^{6+}$ peak shifted to a little lower position, which is probably due to the interaction between single atom Ni and the reconstructed substrate. The morphology of double-shelled hollow structure was not obviously changed after catalytic reactions (Fig. S28-30). However, in the case of $Ni_{SA}$-O/$Mo_2C$ after HER, the enlarged HRTEM image shows original large $Mo_2C$ nanoparticles were crushed into smaller ones and two different lattices of 0.22 nm and 0.23 nm correspond to the d-spacing of (101) and (060) planes of the $Mo_2C$ and $MoO_3$, respectively (Fig. 5a, b). Besides, in the $Ni_{SA}$-O/$Mo_2C$ after OER, the image exhibits the big $Mo_2C$ nanoparticles were changed obviously to smaller ones and the interplanar spacings of 0.21 and 0.23 nm correspond to the (141) and (060) plane of $MoO_3$. These results indicate that $Ni_{SA}$-O/$Mo_2C$ undergoes reconstitution in the process of HER and OER. So, in order to determine the real active species involved in these two reactions, ex-situ XANES and FT-EXAFS spectra after CV and i-t chronoamperometric test for HER and OER were further recorded. As displayed in Fig. 5c, the Mo K-edge XANES spectra for catalysts after HER and OER activation show both positive shifts in the absorption edge compared with that of $Ni_{SA}$-O/$Mo_2C$, indicating the valences Mo oxidation states were increased. However, for catalysts after OER, the Mo K-edge XANES spectra show more positive shifts compared with that of catalysts after HER, indicating catalysts after OER has higher valence state of Mo, which consistent with XPS results. In addition, a new notable shoulder peak in the pre-edge region of OER were shown, suggesting the formation of strongly distorted $MoO_6$ octahedra. Figure 5d,e show the Mo $k^3$-weighted FT-EXAFS spectra of $Ni_{SA}$-O/$Mo_2C$ after HER and OER, respectively. After HER, it shows the main peak centered at ~2.6 Å, corresponding to the Mo-Mo scattering in $Mo_2C$. In particular, two smaller peaks located at around 1.3 Å and 0.88 Å can be found, which are typical characteristic of Mo-O bonds, indicating the surface of $Mo_2C$ was further oxidized[39,40]. In contrast, after OER, the spectra display the existence of the Mo-O bond but no Mo-Mo bond, and its shape is similar with that of molybdate, indicting, except for crystal $MoO_3$, amorphous or low crystallinity molybdate was also formed followed by the possible reaction of $MoO_3$ + $OH^{-2}$= $MoO_4^{2-}$ +$H_2O$[43-45]. In order to clary whether the changes of $Ni_{SA}$-O/$Mo_2C$ is automatic in KOH KOH electrolyte at open circuit potential (OCP). As shown in Fig. conditions, we tested the XRD, Mo k-edge XANES and FT-EXAFS curve for $Ni_{SA}$-O/$Mo_2C$ electrocatalysts after being dipped into KOH electrolyte at open circuit potential (OCP). As shown in Fig. S26b, the $Ni_{SA}$-O/$Mo_2C$ electrocatalysts after being dipped into KOH electrolyte at OCP ($Ni_{SA}$-O/$Mo_2C$-OCP) has similar characteristic peaks with $Ni_{SA}$-O/$Mo_2C$ after HER except the a little weaker $MoO_3$ and $MoO_4^{2-}$ peaks. Based on the dissolution of $MoO_x$ in KOH[43-45], we speculate the original surface $MoO_x$ reacts with KOH to form $MoO_4^{2-}$, and with the consumption of surface $MoO_x$, $Mo_2C$ surface is continuously oxidized to form new surface $MoO_x$ until the $Mo_2C$ is completely dissolved in KOH. However, compared with $Ni_{SA}$-O/$Mo_2C$ after OER, $Ni_{SA}$-O/

$Mo_2C$-OCP has characteristic peaks of $Mo_2C$. In addition, compared with $Ni_{SA}$-O/$Mo_2C$ and $Ni_{SA}$-O/$Mo_2C$ after HER, the absorption edge of $Ni_{SA}$-O/$Mo_2C$-OCP was higher, suggesting its average oxidation state of Mo was more positive. Meanwhile, the Mo k-edge XANES shape and the Mo k3-weighted FT-EXAFS spectra of $Ni_{SA}$-O/$Mo_2C$-OCP are different from $Ni_{SA}$-O/$Mo_2C$ after HER but similar with that of $Ni_{SA}$-O/$Mo_2C$ after OER. These results indicate the dissolution and oxidation of $Mo_2C$ in KOH is automatic in alkaline conditions. The applied oxidation potential of OER process will intensify these two continuous reaction processes, while HER process will inhibit to a certain extent the oxidation of the inner layer of $Mo_2C$ and then partially prevent the formation of $MoO_4^{2-}$ from the outer layer of $MoO_x$. Based on the dissolution of $MoO_x$ in KOH, Fig. 5f, g present comparisons of $Ni_{SA}$-O/$Mo_2C$ catalysts in Ni K-edge XANES and FT-EXAFS after HER and OER. In the case of HER (Fig. 5f), the pre-edge position of XANES spectra show a slight shift to higher energy values compared with the original $Ni_{SA}$-O/$Mo_2C$, indicating that the Ni oxidation state slightly increases but still between $Ni^0$ and $Ni^{2+}$. After carefully compared the FT-EXAFS spectra of $Ni_{SA}$-O/$Mo_2C$ before and after HER, we found the peak of Ni-O did not change, however the position of Ni-O-Mo was shift to higher R, which due to the reconstitution of $Mo_2C$ substrate. After EXAFS fitting, we found the distance of Ni-O (2.05 Å) is almost the same as that of $Ni_{SA}$-O/$Mo_2C$ (2.01 Å) and Ni-Mo (Ni-O-Mo) is increased from 2.40 to 3.31 Å (Table S7). Moreover, the Ni-O coordination number increased to 5.90 while the Ni-Mo decreased to 4.50. And no typical peaks for Ni-Ni bonds are observed as well. These results indicated the Ni was still exists as single atoms while its coordination structure had changed slightly. This catalyst after HER was collectively denoted as $Ni_{SA}$-$MoO_x$/$Mo_2C$. In contrast, after OER (Fig. 5g), the position of the absorption edge is higher than that of the $Ni(OH)_2$ reference, indicating the Ni has the higher oxidation than +2, which probably due to the existence of NiOOH or the strong electron interaction between $Ni(OH)_2$ and the reconstructed substrate (Fig. S31). For the FT-EXAFS spectra of the Ni K-edge after OER, these shapes match well with that of $Ni(OH)_2$. In addition, the Ni-O-Mo coordinated peak almost disappears and new distinct scattering of Ni-Ni emerges at ~2.7 Å. The corresponding fitting EXAFS results display the bond distances Ni-O if 2.05 Å, Ni-Ni (Ni-O-Ni) is 3.07 Å and Ni-Mo (Ni-O-Mo) is 3.34 Å, respectively. The HAADF-STEM images (Fig. 5h) demonstrate the (060) plane of crystal $MoO_3$ and a smaller nanocluster of Ni (yellow circles). In addition, the selected-area elemental mappings at an atomic level further confirm the uniformly distributed Ni and uneven distribution Mo on the surface. This catalyst after OER was collectively denoted as $Ni_{cluster}$/$MoO_x$. These results indicated, after OER, uniformly distributed Ni in $Ni_{SA}$-O/$Mo_2C$ undergoes transformation of the configuration and has been agglomerated into very small clusters by applied potential.

## DFT simulations

In order to further explore the influence of various bonding forms on the electrocatalytic properties of HER and OER, we first analyzed the charge density difference and bader charge. Importantly, all adsorption energy calculations are obtained on the basis of $Ni_{SA}$-O/$Mo_2C$ correction to zero. As shown in the inset of Fig. 6a and Fig. S32, there is a tendency of transferring electrons from Ni site to adsorbates, and we could see explicitly the number of valence electrons of Ni increased when formed Ni-C bond on $Mo_2C$ facet (Fig. 6b) compared with formed Ni-O bond in $Mo_2C$ facet. Therefore, Ni atoms directly embed the plane (101) of $Mo_2C$ has strong adsorption of protons and hydroxyl, while the surface incorporated O of $Mo_2C$ would adjust the number of valence electrons of Ni, then weakening the adsorption capacity. So, the adsorption energies of $Ni_{SA}$-O/$Mo_2C$ for proton and hydroxyl are much lower than those of $Ni_{SA}$-$Mo_2C$, which markedly reduces the likelihood of blocking active sites, thus improving the electrocatalytic properties of HER (Fig. 6a). In addition, we found the energy level of the d-band center of $Ni_{SA}$-O/$Mo_2C$ (−1.66 eV) is much lower than that of

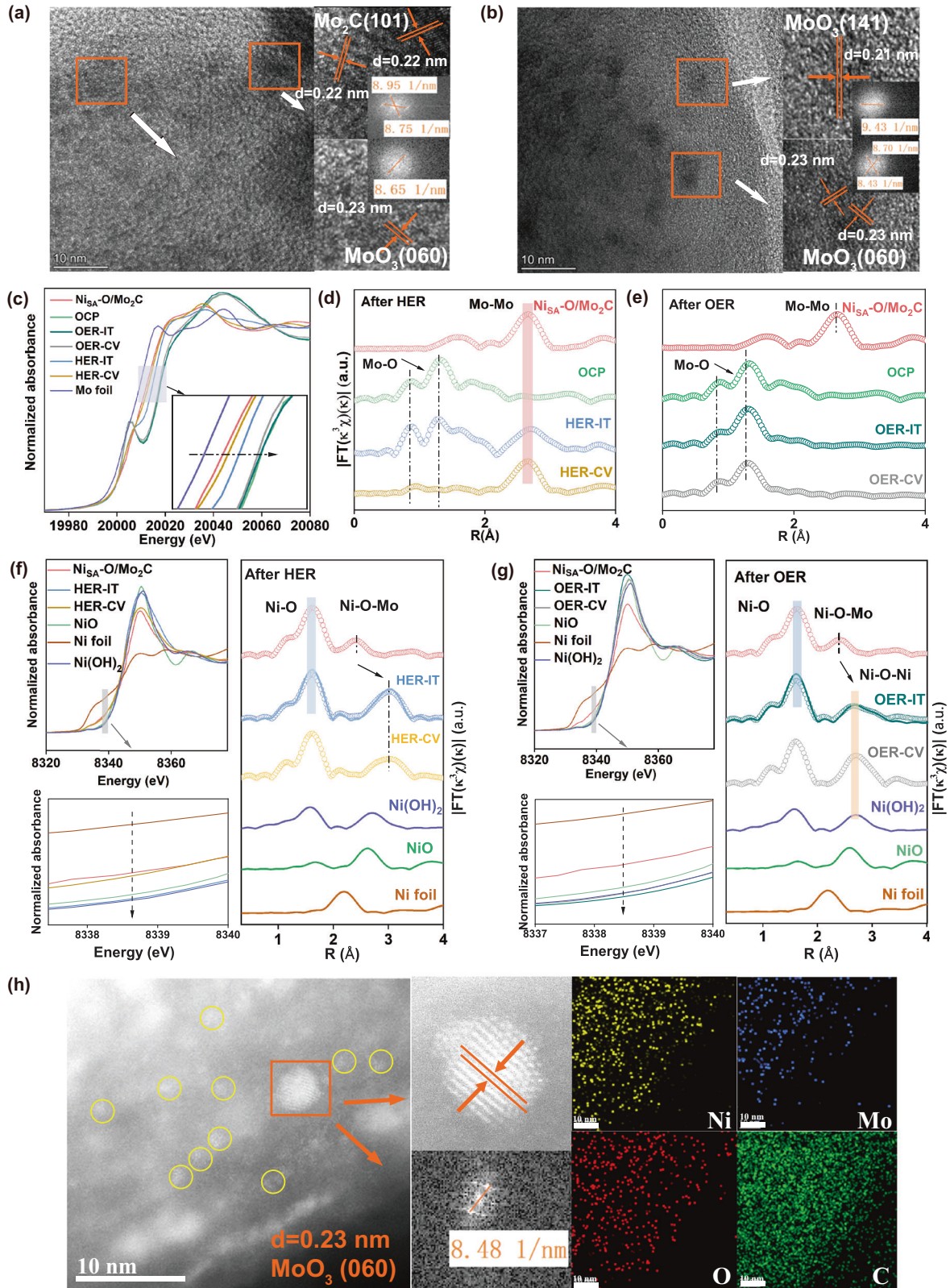

**Fig. 5 | Structural analysis of the Ni_{SA}-O/Mo₂C after HER/OER to study active sites. a, b** HRTEM images of Ni_{SA}-O/Mo₂C after HER/OER stability tests. **c–e** Ex situ Mo k-edge XANES and the corresponding FT-EXAFS curve of Ni_{SA}-O/Mo₂C, Ni_{SA}-O/Mo₂C-OCP, and Ni_{SA}-O/Mo₂C after HER/OER. **f–g** Ex situ XAS spectra of Ni K-edge of Ni_{SA}-O/Mo₂C and the corresponding FT-EXAFS spectra after HER/OER, respectively. **h** HAADF-STEM image and EDS element mapping of Ni_{SA}-O/Mo₂C after OER.

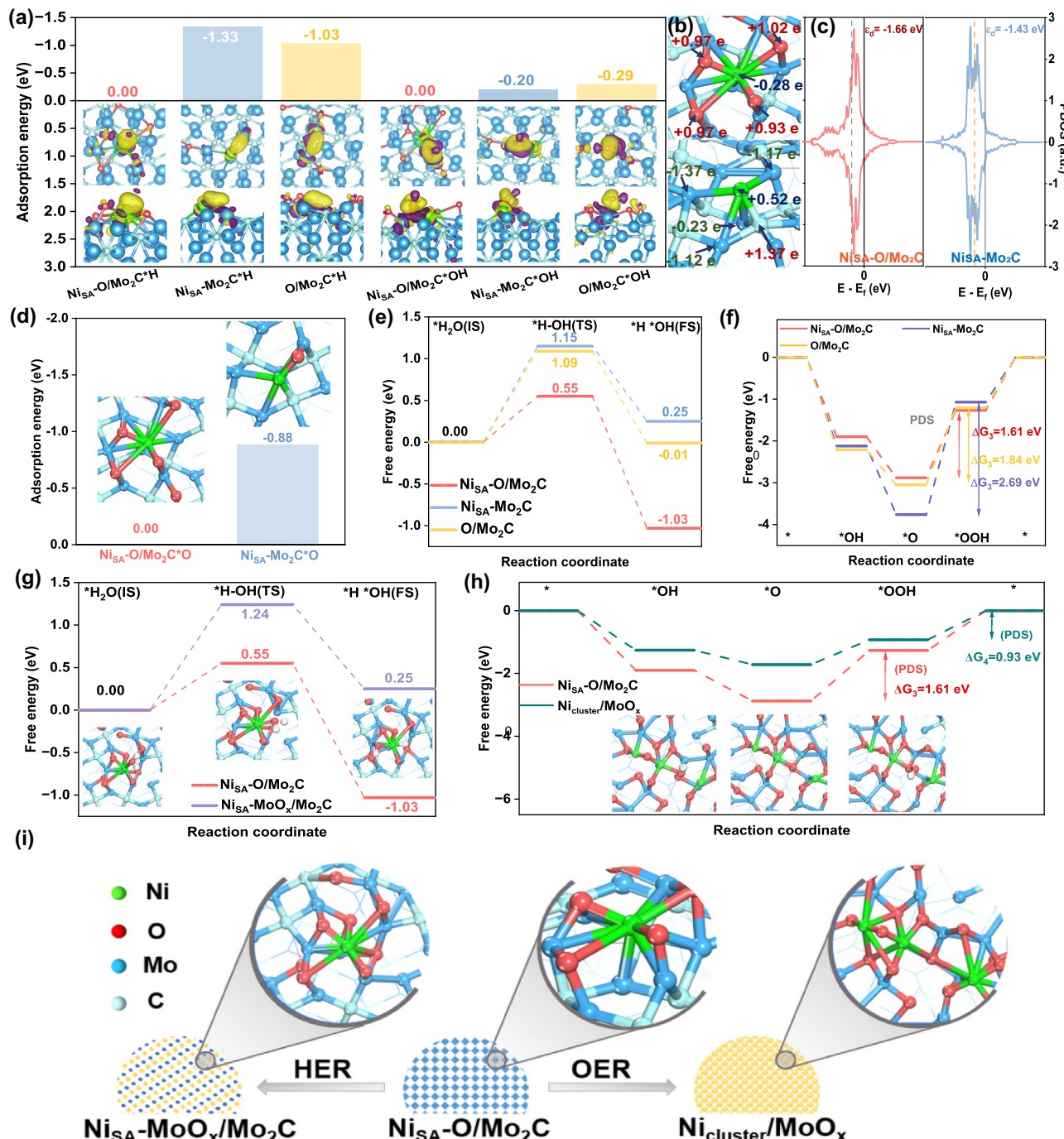

**Fig. 6 | DFT calculation of HER and OER. a** The chemisorption energies of *H and *OH in Ni sites of $Ni_{SA}$-$O/Mo_2C$, $Ni_{SA}$-$Mo_2C$ and $O/Mo_2C$. Insets are their 3D plots of charge density differences: top view and side view. **b** The number of bader charges transfer at specified Ni, O, and Mo atoms in $Ni_{SA}$-$O/Mo_2C$ and $Ni_{SA}$-$Mo_2C$. **c** PDOS graphs of *3d* orbitals of Ni atoms in $Ni_{SA}$-$O/Mo_2C$ and $Ni_{SA}$-$Mo_2C$. The black dash lines show the d-band center ($\varepsilon$), and the Fermi level was taken as zero of energy. **d** The chemisorption energies of O in Ni sites of $Ni_{SA}$-$O/Mo_2C$ and $Ni_{SA}$-$Mo_2C$. **e, g** Gibbs free energy diagram of Volmer step. Inset in (**g**): the optimal models of different stage of $Ni_{SA}$-$MoO_x/Mo_2C$. **f, h** Gibbs free energy change of OER pathways on Ni sites. Inset in (**h**): the atomic configurations of different stage of $Ni_{cluster}/MoO_x$. **i** Structural reconstitution of $Ni_{SA}$-$O/Mo_2C$ catalyst after different reaction process. Water blue= Mo, green = Ni, grey = C, red = O.

$Ni_{SA}$-$Mo_2C$ (−1.43 eV) (Fig. 6c). According to the d-band center theory, the lower electron density near of Fermi level will facilitate the desorption of adsorbate. So, compared with $Ni_{SA}$-$O/Mo_2C$, the $Ni_{SA}$-$Mo_2C$ had much stronger capacity of adsorbed O (Fig. 6d), which will lead to the higher overpotential of OER[46,47]. We then calculated the potential barrier for hydrogen production and the energy barrier for Volmer step, as well as the overpotential of OER. As shown in Fig. S33 and Fig. 6e, the energy of the potential-determining step required by $Ni_{SA}$-$O/Mo_2C$ for hydrogen generation is 0.5 eV, much lower than those of

$Ni_{SA}$-$Mo_2C$ (0.65 eV) and $O/Mo_2C$ (1.18 eV). In addition, the energy barrier of sluggish water dissociation for $Ni_{SA}$-$O/Mo_2C$ is 0.55 eV, which is lower than 1.15 eV of $Ni_{SA}$-$Mo_2C$ and 1.09 eV of $O/Mo_2C$, indicating Ni loaded on partially oxidized $Mo_2C$ can improve HER performance. The processes of Volmer step are shown in Fig. S34. Meanwhile, the Potential Determining Step (PDS) for OER is the third electrochemical step from *O to *OOH, as we can see, $Ni_{SA}$-$O/Mo_2C$ remarkably decreases the energy barrier in this step to 1.61 eV, lower than that of $O/Mo_2C$ (1.84 eV) and $Ni_{SA}$-$Mo_2C$ (2.69 eV) at U=1.23 V, indicating $Ni_{SA}$-

O/Mo$_2$C has much better performances, being consistent with the experimental observations. The diagram of OER reaction step was shown in Fig. S35.

Based on the above results, it would be found that the partially oxidized surface of Mo$_2$C facilitated the binding of Ni atoms via Ni-O and Ni-Mo (Ni-O-Mo) chemical bonds, thus promoting the electrocatalytic properties of HER and OER. However, the O/Mo$_2$C substrate of Ni$_{SA}$-O/Mo$_2$C after HER and OER underwent oxidation and dissolved. For Ni$_{SA}$-MoO$_x$/Mo$_2$C after HER (Fig. 6g, S36), we can see that its energy barrier of water dissociation is 1.24 eV, which becomes larger than that of Ni$_{SA}$-O/Mo$_2$C, indicating the overoxidation of the substrate can lead to some adverse hydrogen evolution effects. But for Ni$_{cluster}$/MoO$_x$ after OER, the PDS changes from the step pf *O to *OOH to the step of *OOH to O$_2$. Its free energy of the rate-determining step is 0.93 eV (Fig. 6h), which becomes smaller than that of Ni$_{SA}$-O/Mo$_2$C at U=1.23 V. In addition, compared with Ni$_{SA}$-O/Mo$_2$C, the reconstructed Ni$_{cluster}$/MoO$_x$ shows a weaker adsorption capacity of O$_2$ (Fig. S37), indicating the reconstructed Ni$_{cluster}$/MoO$_x$ accelerates the oxygen evolution reaction. These results indicate the overoxidation of the substrate is conducive to the aggregation of monatomic Ni into clusters and the improvement of oxygen evolution performance. Figure 6i shows the structural reconstitution of Ni$_{SA}$-O/Mo$_2$C catalyst after different reaction processes. These results are consistent well with the experimental observations.

In summary, we engineered a non-precious Ni single-atom bifunctional catalyst loaded on the surface of oxygen-incorporated Mo$_2$C via Ni-O-Mo bridge bonds (Ni$_{SA}$-O/Mo$_2$C). The obtained Ni$_{SA}$-O/Mo$_2$C catalyst exhibited an overpotential of 133 mV and 299 mV at the current density of 10 mA cm$^{-2}$ for HER and OER, respectively. Comprehensive characterizations and DFT calculations indicated that the surface oxygen atoms of Mo$_2$C were conducive to stabilizing the single atom Ni via the bridge bonding of Ni-O-Mo. The Ni-O-Mo bonding environment results in the regulation of electronic structures of single atom Ni towards an optimized proton and hydroxyl and moderate adsorption energy of O, which determines superior performance of the catalyst. Moreover, we found, that the configuration of atomically dispersed Ni in Ni$_{SA}$-O/Mo$_2$C electrocatalyst underwent transformation and O/Mo$_2$C support were oxidized and dissolved to varying degrees due to instability in KOH and the influence of applied potential. After HER, Ni species still maintain atomically dispersed form. However, after OER, the atomically dispersed Ni were agglomerated into very small clusters. In addition, we found overoxidized Mo$_2$C will lead to the configurations of single atom Ni to reconstruct, which will degrade its HER performance. However, compared with the original Ni$_{SA}$-O/Mo$_2$C, the potential-driven reconstituted small Ni clusters with optimal coordination structures are more favorable for OER. In a word, this work demonstrates a facile and feasible strategy to fabricate a low-cost and high-efficient bifunctional single-atom electrocatalyst and also provides guidance for further design of effective water-splitting electrocatalysts.

## Methods

### Synthesis of amorphous Mo-based precursor

The precursor Mo-based was first synthesized by a one-step hydrothermal method. Heat dissolved 1g of C$_6$H$_{12}$O$_6$ (1.00 g), 0.152 g of CH$_4$N$_2$S, and 0.132 g of C$_{10}$H$_{14}$MoO$_6$ in 50 mL of deionized water, when the solution changed from yellow-green to blue, it was transferred to a 100 mL Teflon-lined autoclave while hot. Then kept in an electric oven at the temperature of 200 °C for 48 h. After the autoclave was cooled down to room temperature, it was centrifuged three times with alternating water and ethanol, respectively. The sample was dried in a vacuum oven at 60 °C for 12 h.

### Synthesis of Ni$_{SA}$-O/Mo$_2$C electrocatalysts

0.20 g of Mo-based precursor and 3 M (21.809 g) of Ni(NO$_3$)$_2$·6H$_2$O were dissolved in 10 mL of deionized water mixed with 15 mL of ethanol. Ultrasonically sonication dissolved for 40 min, constant temperature water bath at 40 °C for one night with continuous stirring. Then washed with ethanol for three times and dried to obtain the surface adsorbed Ni of Mo-based nanospheres. Finally, the sample was annealed at 300 °C for 1 h under an air atmosphere with a heating rate of 2 °C, to ensure that the C in the Mo-based nanospheres structure could escape with air, followed by 750 °C for 3 h under a hydrogen-argon atmosphere at the same rate of warming. The Ni$_{SA}$-O/Mo$_2$C was obtained after annealing.

Similarly, Ni$_{1.4}$-O/Mo$_2$C and Ni$_{4.5}$-O/Mo$_2$C was synthesized with the same method except for the doping amount of nickel nitrate (1.5 M and 4.5 M, respectively).

O/Mo$_2$C was synthesized by direct calcination of Mo-based precursors without Ni(NO$_3$)$_2$·6H$_2$O.

### Structural characterization

The crystalline structure and phase purity were tested by X-ray diffraction pattern (XRD, Ultima, Japan) with Cu Kα radiation at room temperature. The transmission electron microscopy (TEM) measurements were performed on Hitachi H-7650 operated at 2 kV. The Thermo Fisher IRIS Intrepid α system measured ICP-OES, which can be used to determine the element contents of the samples. High-resolution transmission electron microscopy (HR-TEM) and high-angle annular dark-field scanning transmission electron microscopy (HAADF-STEM) were carried out by Talos F200X G2 and JEOL JEM-2100F field emission electron microscope working at 200 kV, respectively. The X-ray photoelectron spectroscopy (XPS) experiments were tested on the PHI 5000 Versaprobe III spectrometer. The Brunauer-Emmett-Teller (BET) specific surface areas of typical products were performed at 77 K in a Belsorp-max surface area detecting instrument. N$_2$ adsorption-desorption experiments were carried out at 77 K on a Micromeritics ASAP 2460 Instrument. X-ray absorption spectroscopy (XAS) measurement and data analysis: X-ray absorption near-edge spectra (XANES) at the Mo K-edge obtained at 1W1B station in BSRF (Beijing Synchrotron Radiation Facility, P. R. China) operated at 2.5 GeV with a maximum current of 250 mA. The C-edge and N-edge were measured at beamline BL12B of the National Synchrotron Radiation Laboratory (NSRL) of China and the samples were deposited onto a double-sided carbon tap.

### Electrochemical measurements

All spectra were collected in ambient conditions. Measurements of HER and OER activity of the electrocatalysts were performed with a traditional three-electrode system (CHI-760E), using a graphitic carbon rod as the counter electrode, Hg/HgO as the reference electrode, and a glassy carbon (GC) or carbon paper with the as-synthesized samples as the working electrode. 3.0 mg of sample and 20 μL of 5 wt% Nafion solution were dispersed in 300 μL of 1:1 v/v water/ethanol, then added 1.5 mg carbon black and sonicated for 30 min to form a homogeneous ink. Then 5 μL of the solution was dispersed on a GC electrode of 3 mm diameter with a loading amount of 0.7 mg cm$^{-2}$. For the OER activity performed, 10.0 mg of sample and 50 μL of 5 wt% Nafion solution in 1mL of 1:1 v/v water/isopropanol, then 80 μL of solution was dispersed on 1×1 carbon paper with loading 0.74 mg cm$^{-2}$. All electrochemical OER and HER experiments were performed in a 1.0 M KOH solution, and polarization curves were iR-corrected. All potentials were referenced to the reversible hydrogen electrode (RHE), following the below equation:

$$E_{RHE} = E_{Hg/HgO} + 0.095 + 0.0591pH \qquad (1)$$

The electrochemically active surface area (ECSA) of the as-synthesized samples without carbon black was estimated by the cyclic voltammetry (CV) measurements[1]. The electrochemical double-layer capacitance was tested from 0.86 to 0.96 V (vs RHE) in 1M KOH solution for the HER or OER process at different scan rates (20, 40, 80, 100). The difference between the anode and cathode currents was plotted linearly against the scan rate, with the slope corresponding to the electrochemical double-layer capacitance ($C_{dl}$).

The ECSA was then estimated using the following equation:

$$ECSA = \frac{C_{dl}}{C_s} \qquad (2)$$

## Density Functional Theory Calculations

A series of density functional theory (DFT) calculations were all done with the Vienna Ab initio Simulation Package (VASP). The spin polarized was considered and calculated with Hubbard model. The electron-ion interaction was described using the projector augmented wave (PAW), and the kinetic energy cutoff for plane wave expansions was set to 450 eV. The electron exchange and correlation energies were treated within a generalized gradient approximation (GGA) in the Perdew-Burke-Ernzerhof (PBE) exchange-correlation. A DFT-D3 scheme of dispersion correction was used to describe the van der Waals (vdW) interactions in molecule adsorption. The Brillouin zone was sampled using the Monkhorst-Pack 2 x 2 x 1 sampling and the convergence criteria were $1 \times 10^{-6}$ eV in structure optimization, and force convergence criterion of $-0.02$ eV/Å. The electron smearing width of $\sigma = 0.05$ eV was employed according to the Methfessel-Paxton technique. To avoid the interactions between two adjacent periodic images, the vacuum thickness was set to be 15 Å, and to simulate the effect of inside a solid, we fixed two-layer atoms at the bottom.

The climbing image-nudged elastic band (CI-NEB) method was used to determine transition states (TS), with the energy convergence criteria of $1 \times 10^{-7}$ eV and force convergence criteria of $-0.03$ eV/Å. The computed vibrational frequencies were used to characterize a minimum state without imaginary frequencies or an authentic transition state with only one imaginary frequency. The kinetic energy barrier ($\Delta G$) of the Volmer step at alkaline HER is applied as an activity descriptor for catalytic performance, which can be calculated as follows:

$$\Delta G = G_{TS} - G_{IS} \qquad (3)$$

where $G_{TS}$ and $G_{IS}$ are the free energy of the transition state and the initial state, respectively.

The free energy correction was obtained by including the zero-point energy (ZPE) and entropic contributions from vibrational degrees of freedom calculated with the substrate fixed, and the value gained by using Vaspkit.1.2.4.

The adsorption energy ($E_{ad}$) was calculated by subtracting the energies of the isolated adsorbate and the catalyst from the total energy of the adsorbed system:

$$E_{ad} = E_{slab+adsorbate} - E_{slab} - E_{adsorbate} \qquad (4)$$

Where $E_{slab+adsorbate}$ is the total energy of adsorbate as adsorbed steadily at the active site. $E_{slab}$ and $E_{adsorbate}$ are the total energy of the adsorbed surface and isolated adsorbate, respectively.

In alkaline media (pH = 14), the four-electron OER pathway may proceed through the following elementary steps:

$$OH^- + * \rightarrow OH^* + e^- \qquad (5)$$

$$OH^* + OH^- \rightarrow O^* + H_2O + e^- \qquad (6)$$

$$O^* + OH^- \rightarrow OOH^* + e^- \qquad (7)$$

$$OOH^* + OH^- \rightarrow O_2^* + H_2O + e^- \qquad (8)$$

$$O_2^* \rightarrow O_2 + e \qquad (9)$$

Where the *refers to the catalytic, and the *one refers to the species that adsorbed on the activity sites.

Neglect PV contribution to translation for adsorbed molecules, the free energy of every step was calculated according to the equation of $G = E + H_{cor} - TS = E + G_{cor}$, where E is the energy of every species obtained from DFT calculations, and S are entropy, while T is 298.15 K. The $H_{cor}$ and $G_{cor}$ are the thermal corrections to enthalpy and the thermal correction to Gibbs free energy, respectively. These $G_{cor}$ of *H, *O, *OH, and *OOH were taken from the frequency DFT calculation and got value by using Vaspkit.1.2.4.

The Gibbs free energy of the proton-electron pairs ($OH^- - e^- = H_2O - (H^+ + e^-)$) related in the PECT progress, whereas the fact that the proton-electron pairs is in equilibrium with gaseous $H_2$ at 0 V versus standard hydrogen electrode (U = 0, pH = 0, and pressure = 1 bar, and temperature = 298.15 K): $\mu(H^+ + e^-) = 1/2\ \mu(H_2\ (g))$. According to Vaspkit.1.2.4, the internal energy of gas molecular gained from the formula: $U(T) = ZPE + \Delta U(0\text{-}T)$, the enthalpy of gas molecular gained from the formula: $H(T) = U(T) + PV = ZPE + \Delta U(0\text{-}T) + PV$, and the Gibbs free energy of gas molecular gained from the formula: $G(T) = H(T) - TS = ZPE + \Delta U(0\text{-}T) + PV - TS = E\_DFT + G\_cor'$. Where E_DFT is the energy of the free gas molecule obtained from DFT calculations, G_cor' is the thermal correction to Gibbs free energy of the free gas molecule obtained from the frequency DFT calculation and got value by using Vaspkit.1.2.4, with the temperature of 298.15 K, the pressure of $H_2O$ and $H_2$ were 0.035 atm and 1 atm, respectively, and both $H_2O$ and $H_2$ input 1 as the value of spin multiplicity. Note that the free energy of an $O_2$ gas molecule should be calculated by this equation: $G(O_2, g) = 2*G (H_2O, g) - 2*G (H_2, g) + 4.92$ eV.

The d-band center of the 3d orbitals of Ni sites obtained from their PDOS by using equation:

$$\varepsilon_d = \frac{\int_{-\infty}^{+\infty} \varepsilon f(\varepsilon) d\varepsilon}{\int_{-\infty}^{+\infty} f(\varepsilon) d\varepsilon} \qquad (10)$$

where, the $f(\varepsilon)$ is the PDOS of an energy level of $\varepsilon$.

The charge density difference was evaluated using the formula $\Delta\rho = \rho(\text{substrate + adsorbate}) - \rho(\text{adsorbate}) - \rho(\text{substrate})$, then analyzed by using the VESTA code.

## Construction of structure

**$Mo_2C$.** A 2x3 supercell surface consisting of 144 atoms from a cleaving surface of plane (1 0 1) of orthorhombic $Mo_2C$ primitive cell was used firstly, and the fractional thickness set to 2.0, then adding 15 Å vacuum thickness to build vacuum slab crystal. The Brillouin zone integration is performed using the uniformly distributed scattering of going through the Gamma point to select a 2x2x1 k-mesh in the Monkhorst-Pack grid to make structure optimization, and we select 3x3x1 k-mesh to do density of state (DOS) calculations.

**$O/Mo_2C$.** A partially oxidized surface of $Mo_2C$ is constructed by loading six oxygen atoms at the most stable adsorption site on the surface of $Mo_2C$.

$Ni_{SA}$-$Mo_2C$: A Ni atom coordinating with C atom is loaded in $Mo_2C$-$O_6$ surface.

**Ni$_{SA}$-Mo$_2$C.** The Mo$_2$C surface is loaded with more oxygen atoms to simulate a further oxidized surface, and then a nickel atom coordinating with four oxygen atoms is loaded on the surface.

**Ni$_{SA}$-MoO$_x$/Mo$_2$C.** The Mo$_2$C surface is loaded with more oxygen atoms to simulate a further oxidized surface and some of the carbon atoms are replaced by oxygen atoms to form a partially disordered MoO$_x$ surface, then a nickel atom coordinating with six oxygen atoms is loaded on the surface.

**Ni$_{cluster}$/MoO$_x$.** The Mo$_2$C surface is loaded with more oxygen atoms to simulate a further oxidized surface and some of the carbon atoms are replaced by oxygen atoms to form a partially disordered MoO$_x$ surface, then three nickel atoms coordinating with six oxygen atoms, respectively, are loaded on the surface.

To simulate the effect inside a solid, the atoms located in the lower half of the slab were fixed for all calculation models.

## Data availability
The data that support the findings of this study are available from the corresponding author upon reasonable request.

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

## Acknowledgements

This work was supported by the National Key R&D Program of China (2023YFB4005100), National Natural Science Foundation of China (22309011, 52272186, 22105116, 21872008, 52173232, 21925202, U22B2071), Yunnan Provincial Science and Technology Project at Southwest United Graduate School (202302AO370017), International Joint Mission On Climate Change and Carbon Neutrality, Beijing Institute of Technology Research Fund Program for Young Scholars. We thank the 1W1B station for XAFS measurements in the Beijing Synchrotron Radiation Facility (BSRF) and Dr. Fang Zhang from Analysis & Testing center, Beijing Institute of Technology.

## Author contributions

D.Z., J.T.Z., M.H.C., and C.C. supervised the project. M.Y.H. and D.Z. designed the work and carried out most of the experiments. L. R. Z. guided the XANES, ex situ XAS measurements, and analyzed XAS results. W.Y.F., X.T., J.T. F,T. X.W. helped to prepare samples and character the samples with TEM and XPS. All the authors discussed the results and assisted during the manuscript preparation.

## Competing interests

The authors declare no competing interests.
