## [Peer Review File · Nature Communications]

Microenvironment Reconstitution of Highly Active Ni Single Atoms on Oxygen-Incorporated Mo₂C for Water SplittingReviewers' Comments:

Reviewer #1:

Remarks to the Author:

Comments

The manuscript reports the development of single-atom-nickel-linked Mo₂C-based bi-functional water-splitting catalysts. Single-atom Ni-linking via surface oxygen to Mo₂C leads to significant improvement over HER and OER catalytic efficiencies. Although the overpotentials, 133 mV and 299 mV for HER and OER at 10 mA per cm²) respectively, are significantly higher as compared to the state-of-the-art non-noble catalysts but the manuscript provided much scientific insight into the dynamical phase evolution during catalysis. However, there are some major concerns that need to be addressed before further consideration

(1) The distribution of Mo and C is uniform through the particle (Figure S2) of the precursor, however, it changes after pyrolysis. The distribution of Mo is certainly not uniform in Mo₂C, while C is uniform as is clear from Figure 1d.

If the particle is pure phase molybdenum carbide as claimed, the distribution of Mo and C should be similar, which is not the case here. This indicates that the sphere in Figure 1d may be a carbon sphere containing Mo₂C, which is generally expected during the pyrolysis of Mo-containing organic precursors [Adv. Funct. Mater. 29 (2019)1807419].

The XPS spectrum of C1s (Figure S8) having C=C and oxygenated functional groups also suggests that the synthesized sample may be a carbon sphere containing Mo₂C. Further, the presence of a uniform carbon signal in the elemental mapping (Figure 5h) after OER (after the full conversion of Mo₂C into oxide) also indicates the sphere is made of carbon.

However, there is neither any mention of such carbon structure in this manuscript nor any discussion of XPS of C1s was made. The content of Mo and C in the sample may be checked from XPS whether it corresponds to a 2:1 ratio, or whether there is excess carbon coming from such a carbon sphere. Further, HRTEM in the different regions of the sphere (darker and lighter regions of 1c) can also be checked to see whether the lighter regions are carbon structures only.

(2) It is better to incorporate the mechanism of core-shell structure formation during Mo₂C formation.

(3) The positions of the major peaks of the catalyst-after-OER do not match with the reference spectrum provided, all the peaks are shifted as compared to their expected positions. The origin of the same should be explained.

(4) What is the source of the surface oxide layer in the as-synthesized Mo₂C? Is this due to unintentional oxidation of Mo₂C during synthesis? How controlled is the oxidation level? The controllability and reproducibility of the oxide layer are important as nickel single atom is essentially linked with such oxygen. Further, why the intensity and area under the curves of Mo⁴⁺ and Mo⁶⁺ (related to surface MoO_x) are so high as compared to Mo²⁺ (related to Mo₂C) despite the sample being predominantly Mo₂C phase?

(5) The performance of HER and OER depends on both density of active sites and the intrinsic activity of those sites. Significant variation in ECSA indicates that there is a variation in the number of active sites among the samples. To clarify which sample has the best intrinsic activity ECSA normalized LSV curves for both HER and OER need to be presented.

(6) Figure 5b shows that the Tafel slope of Ni_{4.5}/Mo₂C is the lowest (89.36 mV/Dec), however, in the text the same slope was assigned to NiSA-O-Mo₂C. Please check the same.

(7) The oxidation of Mo₂C during OER is accountable as the catalytic surface can get oxidized via generated oxygen. But, what is the origin of oxidation of the catalysts during HER? It can be checked

whether such oxidation happens spontaneously in alkaline conditions when the catalysts were dipped into the KOH electrolyte without doing any HER. This will clarify whether the oxidation of Mo₂C is automatic in alkaline conditions or is induced by the HER process.

(8) The surface of the catalysts after both HER and OER changes into MoO_x. However, it is well documented that molybdenum-based oxides are unstable in alkaline conditions [Trans. Nonferrous Met. Soc. China 28(2018) 177], and in fact, Molybdenum based chalcogenides are known to etch out through anodic oxidation process during OER process [Nature Communications 13 (2022) 2191]. It raises the question of whether the catalyst possesses long-term stability as the data for only 15 hours is given. Long-term stability of at least 100 hours needs to be performed to check the stability of the catalyst and emphasize its importance in the field of water splitting.

(9) The NiSA-O-Mo₂C is a pre-catalyst that undergoes oxidation, and the resultant Ni-Cluster-MoO_x is the main OER catalyst. Thus, in my view, the discussion of the high OER activity of NiSA-O-Mo₂C based on the d-band center is not suitable (page 8). The discussion of Ni-Cluster-MoO_x which was made in the later part may be sufficient and suitable.

(10) As per the EXAFS result, the coordination number of nickel (for Ni-O path) changes from 4 to 6 after both HER and OER, however, nickel is coordinated with less number of oxygen atoms in the DFT model ((Figure 6g). DFT should be done taking into consideration of same coordination environment as revealed by EXAFS.

(11) There are some general comments as well. The XANES and TEM were done before and after catalysis, however, in the abstract, it is mentioned that in-situ measurements were done, which is not true. In several sections the performance of the samples was overemphasized, for example, NiSA-O-Mo₂C requires 1.64 V for an overall water-splitting current density of 10 mA/cm², which is only 0.05 V less as compared to Pt/C-IrO₂ couple, which should not be called 'much better performance', it is comparable only. Similarly, it has been mentioned in the abstract section that the HER performance of NiSA-O-Mo₂C is 'comparable to Pt/C' which is not the case here, there is huge difference in their overpotentials as per Figure 4c.

There are certain typo errors like 'nationally designed' in the abstract section, and 'banding energy' in the XPS section. So necessary correction should be done.

Reviewer #2:

Remarks to the Author:

Chen and co-workers developed a high performance non-noble-metal bifunctional catalyst with single Ni atom supported on oxygen-incorporated Mo₂C via Ni-O-Mo bridge bonds. They for the first time found some interesting results. In different reactions, the atomic Ni underwent different microenvironment reconstitution. After HER, Ni species existed in atomically dispersed form. However, after OER, the atomically dispersed Ni were agglomerated into very small clusters by applied potential. So, the work is a nice discussion on the activity of single atom Ni on oxygen-incorporated Mo₂C starting from unprecedented synthesis, characterization electrocatalytic reactions and followed by the revelation of the post-reaction states of Ni. The discussion is valuable and easy to follow. Hence the manuscript is strongly proposed for acceptance in Nature Communications after considering the following minor revisions:

1. The main sample is single Ni atom supported on oxygen-incorporated Mo₂C. The support Mo₂C were partially oxidized or oxygen-incorporated. How about the samples of oxygen-incorporated Ni_{1.5}/Mo₂C, Ni_{4.5}/Mo₂C and pure Mo₂C? Whether there is also partial oxidized in the surface of them? Please provide the data, such as Mo 3d XPS.

In addition, if their surface were also oxygen-incorporated, I recommend the names of the three control samples should be change to Ni_{1.5}-O/Mo₂C, Ni_{4.5}-O/Mo₂C and O/Mo₂C, respectively.

2. How about the Ni 2p XPS spectra of Ni_{1.5}/Mo₂C catalyst?
3. Figure S4 showed the N₂ adsorption-desorption isotherms, the specific areas increased after loading Ni in the surface, please explain this phenomenon.
4. In Figure S1, the amorphous Mo-based precursor didn't look like smooth surface, the authors should provide accurate description.
5. In Figure 5d-e, the disappearance of Mo-Mo diffraction peak after OER can be obviously observed. The author pointed out that it may be due to the further reaction between MoO₃ and KOH. Why did HER remain in MoO₃ state?
6. There are no pages in Ref. 15, 16, 20, 22 and 23, please check the whole ref. carefully.
7. Scale bar for the inset in Fig 6c should be shown.
8. In Figure 4d, the significant numbers are inconsistent. In addition, "The cell are 1.69 V, 1.855 V and 1.9 V, while Pt/C||IrO₂ requires 1.64 V, 1.91V and 1.93V, respectively.
9. Recently some reported single atom catalysts for electrocatalysis should be cited, such as *Advanced Materials*, 2023, <https://doi.org/10.1002/adma.202303243>; *Adv Funct Mater*, 2023, 2210867; *ACS Catalysis*, 2022, 12, 10771–10780.

Reviewer #4:

Remarks to the Author:

In this work, the author designed and fabricated non-noble-metal bifunctional catalysts with single Ni atom supported on oxygen-incorporated Mo₂C via Ni-O-Mo bridge bonds. The HER and OER performance is analysed with DFT calculation and various experiments. However it suffers from the follows problems and cannot be published in *Nature Communications*.

(1) The electronic structure characterisation and calculation did not mention the spin state of Ni. It is well known that Ni has difference spin states and this is critical to understand its electronic structure, especially for oxides. Ni-O bonding cannot be well describe by PBE so I would suggest the author consider the spin polarised calculation with Hubbard model.

(2) The authors claim that "Ni-O-Mo bonding should dominate in the surface" (consistent with TEM image as in Figure 1), while in the atomic structure model only one Ni is placed in the supercell. The author did not provide the information about the supercell so I cannot estimate the Ni concentration but it is obvious far from that in the TEM image.

(3) In Figure 1 d, the scale bar is missing. Is it the same particle as in Figure 1 c? Besides, the distribution of C is significantly different from other elements. The authors did not provide any explanation for this.

(4) The DFT simulation just considered the surface of MoC₂, without the top layer of oxides which is different from the TEM images. Is it a free surface as in DFT models or an interface as in TEM images?

(5) The authors conclude that OER leads to Ni cluster while HER needs Ni single atom as catalytic site, which means it cannot be bifunctional at the same time or alternatively.

(6) There are lots of typos, such as 'nationally designed' in the abstract.

(7) The Ni single atom catalyst's HER performance (the overpotential etc.) is not as good as Ni-doping in Mo₂C. Check *Applied Catalysis B: Environmental* 307, 121201, where Ni-doped Mo₂C gives a better performance.

(8) The final state in Figure 6 (e) is missing. Why are they different from each other?

Response to Reviewers' Comments and Revisions Made

Response to Reviewer #1's Comments:

The manuscript reports the development of single-atom-nickel-linked Mo₂C-based bifunctional water-splitting catalysts. Single-atom Ni-linking via surface oxygen to Mo₂C leads to significant improvement over HER and OER catalytic efficiencies. Although the overpotentials, 133 mV and 299 mV for HER and OER at 10 mA per cm²) respectively, are significantly higher as compared to the state-of-the-art non-noble catalysts but the manuscript provided much scientific insight into the dynamical phase evolution during catalysis. However, there are some major concerns that need to be addressed before further consideration

Response:

We are very pleased that the reviewers have recognized our manuscript in terms of importance and novelty. Special thanks to the reviewers for insightful comments, so that we can further improve the quality of the manuscript.

(1) The distribution of Mo and C is uniform through the particle (Figure S2) of the precursor, however, it changes after pyrolysis. The distribution of Mo is certainly not uniform in Mo₂C, while C is uniform as is clear from Figure 1d. If the particle is pure phase molybdenum carbide as claimed, the distribution of Mo and C should be similar, which is not the case here. This indicates that the sphere in Figure 1d may be a carbon sphere containing Mo₂C, which is generally expected during the pyrolysis of Mo-containing organic precursors [Adv. Funct. Mater. 29 (2019)1807419]. The XPS spectrum of C1s (Figure S8) having C=C and oxygenated functional groups also suggests that the synthesized sample may be a carbon sphere containing Mo₂C. Further, the presence of a uniform carbon signal in the elemental mapping (Figure 5h) after OER (after the full conversion of Mo₂C into oxide) also indicates the sphere is made of carbon. However, there is neither any mention of such carbon structure in this manuscript nor any discussion of XPS of C1s was made. The content of Mo and C in the sample may be checked from XPS whether it corresponds to a 2:1 ratio, or whether there is excess carbon coming from such a carbon sphere. Further, HRTEM in the different regions of the sphere (darker and lighter regions of 1c) can also be checked to see whether the lighter regions are carbon structures only.

Response:

Thank you for the valuable comment. According to the reviewer's good suggestion, we have checked the content of Mo and C in the Ni_{SA}-O/Mo₂C from XPS. As shown in Table S2, it is found that the content of Mo was 10.08 % while the content of C was 45.9 %, which was much higher than the ratio of 2:1. Meanwhile, as shown in the high-resolution TEM image in Fig. S10, the carbon substrate is observed in the lighter regions of the surface of sphere (highlighted by yellow circles), while the darker regions are Mo₂C nanocrystals with lattice stripes. In addition, we also check the elemental mapping of Mo, Ni, O, C in a Ni_{SA}-O/Mo₂C sphere of Fig. 1d. We found, indeed, compared with Mo, the distribution of C is uniform. Meanwhile, the XPS spectrum of C1s in Fig. S14a had C=C and oxygenated functional groups. The XPS spectrum of C1s (Fig. S14a) is fitted into four different signals at 283.9, 284.8, 285.8 and 288.9 eV,

which are attributed to C-Mo, C-C/C=C, C-O, and O-C=O, respectively. In Fig. 5h, after OER, the carbon signal is uniform as well with the full conversion of Mo₂C into oxide. So, just as the reviewer said, the synthesized sample is a carbon sphere containing Mo₂C, which is generally expected during the pyrolysis of Mo-containing organic precursors.^{r1, r2} From the XRD patterns of Ni_{SA}-O/Mo₂C, these diffraction peaks of Ni_{SA}-O/Mo₂C are obvious, indicating the content of Mo₂C is high and the crystallinity is good. So, compared with carbon support, we consider the performance Ni_{SA}-O/Mo₂C spheres mainly come from Mo₂C phase. However, the carbon substrate can stabilize the Mo₂C nanocrystals and provide better electrical conductivity. We also added the discussion of XPS of C1s in **Supplementary Information**.

Revision:

Left column, Page 3

“During the pyrolysis process, the Mo-based organic precursors were simultaneously converted to Mo₂C nanocrystals with Ni ions fixed on their surfaces, which embedded in a uniform carbon substrate (Fig. S1).”

Right column, Page 3

“Energy-dispersive X-ray spectroscopy (EDS) mapping images reveal that Mo, Ni and O elements almost have the same double-shelled hollow shape and C elements is uniformly distributed in an individual Ni_{SA}-O/Mo₂C nanospheres (insert in Fig. 1c, and Fig. 1d). Combined with the HR-TEM magnified image in Fig. S10, we further confirm that Mo₂C nanocrystals with Ni ions were embedded in a uniform carbon substrate.”

Left column, Page 4

“The XPS spectrum of C1s (Fig. S14a) is fitted into four different signals at 283.9, 284.8, 285.8 and 288.9 eV, which are attributed to C-Mo, C-C/C=C, C-O, and O-C=O, respectively.”

Fig. S10, page 10

Fig. S10 HR-TEM images of the Ni_{SA}-O/Mo₂C.

Fig. S14, page 11

Fig. S14a XPS spectrum of the O-Mo₂C and Ni_{SA}-O/Mo₂C in the C 1s, respectively.

Table S2, page 22

Table S2 The content of each element in the Ni_{SA}-O/Mo₂C calculated by XPS.

Name	Mo	Ni	C	O
Atomic (%)	10.08	0.91	45.9	15

References

35. H. Wei, Q. Xi, X. A. Chen, D. Guo, F. Ding, Z. Yang, S. Wang, J. Li and S. Huang, *Adv. Sci.*, 2018, **5**, 1700733.

The ref. r1-2

- r1. Z. Wu, J. Wang, R. Liu, K. Xia, C. Xuan, J. Guo, W. Lei and D. Wang, *Nano Energy*, 2017, **32**, 511-519.
- r2. S. Li, C. Cheng, A. Sagaltchik, P. Pachfule, C. Zhao and A. Thomas, *Adv. Funct. Mater.*, 2019, **29**, 1807419.

(2) It is better to incorporate the mechanism of core-shell structure formation during Mo₂C formation.

Response:

We are thankful for the reviewer's constructive comments. Based on the reviewer's comment, we recognized that the description of core-shell structure of Ni_{SA}-O/Mo₂C in our original manuscript may not be accurate, double-shelled hollow structure should be more appropriate. In order to understand the formation of double-shelled hollow structure, we further monitored the evolution of structures along with the increasing pyrolysis temperature. The evolution of pyrolysis intermediates along with temperature was then tracked via TEM and corresponding elemental mapping of Mo, Ni, O, C. We selected four pyrolysis intermediates to monitor the evolution of

structures: low temperature calcination at 300 °C in air; 300 °C in air followed by 450 °C; 300 °C in air followed by 550 °C and 300 °C in air followed by 650 °C for 3 h under hydrogen-argon (H₂/Ar) atmosphere, respectively. Combined with the original Mo-based organic precursors and the main sample obtained under the condition of 300 °C in air followed by 750 for 3 h, we made a comparison of Fig. S3. It was noting that the double-shelled hollow structure of Ni_{ISA}-O/Mo₂C was obtained at certain temperatures under air and sequential H₂/Ar atmospheres. From SAED pattern images (a₁-a₆), all of them well maintain the morphology of nanosphere of the precursor. Mo, Ni and O elements almost have the same shape and C elements is uniformly distributed in an individual Ni_{ISA}-O/Mo₂C nanospheres. The original Mo-based organic precursors are sold. After calcination at 300 °C in air (a₂), the intermediate shows an fuzzy core-shell structure, in which Mo and Ni elements aggregate towards the core. Meanwhile, after XRD tests, the weakly crystalline of Mo-based organic precursors transform to amorphous MoO_x-related material (Fig. S4), which can be proved by the more obvious O element mapping of intermediate a₂. Subsequently, with the elevation of pyrolysis temperatures in H₂/Ar atmosphere, the obvious core-shell structure is formed. Then the core become smaller and smaller, until double shelled Ni_{ISA}-O/Mo₂C hollow nanosphere is gradually formed. The distribution of Mo element is completely consistent with the structural changes. In order to verify the roles of air and H₂/Ar atmospheres. We first calcined the Mo-based organic precursors with only H₂/Ar atmospheres. As shown in Fig. S5, there is no significant changes between the shape of the calcined sample and the precursor, which means that the first step of air calcination is very important in the formation of double-shelled hollow structure. Then, we calcined the Mo-based organic precursors with air and subsequent Ar atmospheres. As shown in Fig. S6, although there is a small amount of core-shell structure, there is no double-shelled hollow structure and the spherical shape of precursor is not maintained very well. Based on all the results above, we conclude the formation of double-shelled hollow structure is mainly based on the early oxidation of air and non-equilibrium non-uniform shrinkage caused by heat treatment. The schematic illustration is shown in Fig. S7. At the initial stage of calcination in air, there exists an early oxidation from air and meanwhile a large temperature gradient (ΔT) along the radial direction, which leads to the formation of a MoO_x and MoO_y fuzzy core-shell structure. Two forces of opposite directions (the contraction force: F_{co} ; the adhesion force: F_{ad}) act on the interface between the MoO_x shell and the MoO_y core (the so-called heterogeneous contraction). Subsequently, during the calcination in H₂/Ar atmospheres, when F_{co} is larger than the F_{ad} , the larger F_{co} causes the inner core to shrink inward, and the F_{ad} of the shell prevents it from shrinking inward, eventually resulting in the core shrinking and separating from the shell. With the increase of heating temperature, F_{co} decreases continuously, and when F_{ad} exceeds F_{co} , the direction of material movement will be reversed. Eventually, the core shrinks outward, leaving a hole in the core. Based on this formation mechanism, the step of air calcination is conducive to the formation of amorphous molybdenum oxide. The H₂/Ar atmosphere helps to form Mo₂C rigid shell that generates F_{ad} and also is conducive to the volatilization of internal substances and enhance F_{co} .

Fig. S3-7, page 6-8

Fig. S3 Comparison of the morphologies and structures of (a₁) Mo-based precursor, (a₂) Ni_{SA}-O/Mo₂C-300 °C air, (a₃) Ni_{SA}-O/Mo₂C-300°C air - 450 H₂/Ar, (a₄) Ni_{SA}-O/Mo₂C-300°C air -550 H₂/Ar, (a₅) Ni_{SA}-O/Mo₂C-300°C air -650 H₂/Ar and (a₆) the Ni_{SA}-O/Mo₂C-300 °C air -750 H₂/Ar. It mainly presents SAED pattern images and corresponding elemental mapping images of Ni, Mo, O and C elements of all samples, respectively.

Fig. S4 XRD patterns of Mo-based precursor, Ni_{SA}-O/Mo₂C-300 °C -air which

pyrolyzed at 300° in air atmosphere for 1h after adsorption of Ni ions.

Fig. S5 HR-TEM image, SAED pattern and EDX mapping of Mo, Ni, C and O elements of Ni_{SA}-O/Mo₂C-H₂/Ar.

Fig. S6 TEM image of Ni_{SA}-O/Mo₂C-air-Ar: Pyrolysis at 300° for 1h in air atmosphere, followed by 750 °C for 3 h under Argon atmosphere with with a heating rate of 2 °C.

Fig. S7 Schematic illustration of the fabrication of double-shelled Ni_{SA}-O/Mo₂C hollow nanosphere.

Revision:

Right column, Page 3

“After pyrolysis, an obvious double-shelled hollow structure can be observed (Fig. 1a, b). The mechanism of double-shelled hollow structure formation has been studied in detail (Fig. S3-S6) and its possible schematic illustration is shown in Fig. S7, which is mainly based on the early oxidation from air and non-equilibrium non-uniform shrinkage caused by heat treatment.^{31,32} During the formation of double-shelled hollow structure, the conditions of air and H₂/Ar atmospheres are both vital important.”

Revision in Supplement information:

Page 6-9

“The evolution of pyrolysis intermediates along with temperature was then tracked via TEM and corresponding elemental mapping of Mo, Ni, O, C. We selected four pyrolysis intermediates to monitor the evolution of structures: low temperature calcination at 300 °C in air; 300 °C in air followed by 450 °C; 300 °C in air followed by 550 °C and 300 °C in air followed by 650 °C for 3 h under hydrogen-argon (H₂/Ar) atmosphere, respectively. Combined with the original Mo-based organic precursors and the main sample obtained under the condition of 300 °C in air followed by 750 for 3 h, we made a comparison of Fig. S3. It was noting that the double-shelled hollow structure of Ni_{SA}-O/Mo₂C was obtained at certain temperatures under air and sequential H₂/Ar atmospheres. From SAED pattern images (a₁-a₆), all of them well maintain the morphology of nanosphere of the precursor. Mo, Ni and O elements almost have the same shape and C elements is uniformly distributed in an individual Ni_{SA}-O/Mo₂C nanospheres. The original Mo-based organic precursors are sold. After calcination at 300 °C in air (a₂), the intermediate shows an fuzzy core-shell structure, in which Mo and Ni elements aggregate towards the core. Meanwhile, after XRD tests, the weakly crystalline of Mo-based organic precursors transform to amorphous MoO_x-related material (Fig. S4), which can be proved by the more obvious O element mapping of intermediate a₂. Subsequently, with the elevation of pyrolysis temperatures in H₂/Ar atmosphere, the obvious core-shell structure is formed. Then the core become smaller and smaller, until double shelled Ni_{SA}-O/Mo₂C hollow nanosphere is gradually formed. The distribution of Mo element is completely consistent with the structural changes. In order to verify the roles of air and H₂/Ar atmospheres. We first calcined the Mo-based organic precursors with only H₂/Ar atmospheres. As shown in Fig. S5, there is no significant changes between the shape of the calcined sample and the precursor, which means that the first step of air calcination is very important in the formation of double-shelled hollow structure. Then, we calcined the Mo-based organic precursors with air and subsequent Ar atmospheres. As shown in Fig. S6, although there is a small amount of core-shell structure, there is no double-shelled hollow structure and the spherical shape of precursor is not maintained very well. Based on all the results above, we conclude the formation of double-shelled hollow structure is mainly based on the early oxidation from air and non-equilibrium non-uniform shrinkage caused by heat treatment. The schematic illustration is shown in Fig. S7. At the initial stage of calcination in air, there exists an early oxidation from air and meanwhile a large

temperature gradient (ΔT) along the radial direction, which leads to the formation of a MoO_x and MoO_y fuzzy core-shell structure. Two forces of opposite directions (the contraction force: F_{co} ; the adhesion force: F_{ad}) act on the interface between the MoO_x shell and the MoO_y core (the so-called heterogeneous contraction). Subsequently, during the calcination in H_2/Ar atmospheres, when F_{co} is larger than the F_{ad} , the larger F_{co} causes the inner core to shrink inward, and the F_{ad} of the shell prevents it from shrinking inward, eventually resulting in the core shrinking and separating from the shell. With the increase of heating temperature, F_{co} decreases continuously, and when F_{ad} exceeds F_{co} , the direction of material movement will be reversed. Eventually, the core shrinks outward, leaving a hole in the core. Based on this formation mechanism, the step of air calcination is conducive to the formation of amorphous molybdenum oxide. The H_2/Ar atmosphere helps to form Mo_2C rigid shell that generates F_{ad} and also is conducive to the volatilization of internal substances and enhance F_{co} .

Added ref. 33-34

33. L. Zhou, D. Zhao and X. W. Lou, *Adv. Mater.*, 2012, **24**, 745-748.

34. J. Guan, F. Mou, Z. Sun and W. Shi, *Chem. Commun.*, 2010, **46**, 6605–6607.

(3) The positions of the major peaks of the catalyst-after-OER do not match with the reference spectrum provided, all the peaks are shifted as compared to their expected positions. The origin of the same should be explained.

Response:

Thank you for the reviewer's good question. Indeed, in the XPS spectra of Mo and Ni 3d, these positions of the major peaks of the catalyst after OER are shifted as compared to their standard samples (Fig. S25 and 29). In Mo 3d spectra of Fig. S25, the high-resolution Mo 3d spectra of pure Mo_2C were first deconvoluted into six peaks, corresponding to Mo^{2+} (228.6 eV, 231.7 eV), Mo^{4+} (229 eV, 232 eV) and Mo^{6+} (232.8 eV, 235.8 eV) species. Just as we found in this work, after HER and OER, the $\text{Ni}_{\text{ISA-O}}/\text{Mo}_2\text{C}$ were oxidized to varying degrees under applied potentials. Compared with HER, the oxidized degree of $\text{Ni}_{\text{ISA-O}}/\text{Mo}_2\text{C}$ after OER was more serious, which will lead to its higher valence states. So, despite they all show major valence states of about averaging 6, we can find that taking the peak position of Mo^{6+} as the standard, the peak of $\text{Ni}_{\text{ISA-O}}/\text{Mo}_2\text{C}$ after HER was a little lower while the peak of $\text{Ni}_{\text{ISA-O}}/\text{Mo}_2\text{C}$ after OER was a little higher. These results are very consistent with the Mo k-edge XANES of $\text{Ni}_{\text{ISA-O}}/\text{Mo}_2\text{C}$ after HER and OER states in Fig. 5c, in which the Mo K-edge XANES spectra for catalysts after HER and OER all show positive shifts in the absorption edge compared with that of $\text{Ni}_{\text{ISA-O}}/\text{Mo}_2\text{C}$. However, for catalysts after OER, it shows more positive shifts compared with that of $\text{Ni}_{\text{ISA-O}}/\text{Mo}_2\text{C}$ after HER, indicating the Mo oxidation states all increased and $\text{Ni}_{\text{ISA-O}}/\text{Mo}_2\text{C}$ after OER increased more. In addition, after OER (Fig. 5g), the position of the absorption edge of Ni K-edge spectra is higher than that of the $\text{Ni}(\text{OH})_2$ reference. And in the spectrum of Ni, when we use $\text{Ni}(\text{OH})_2$ as a standard sample, the Ni peak position of OER is also shifted to a little higher energy, both indicating the Ni has the higher oxidation than +2, which probably due to the

existence of NiOOH or the strong electron interaction between Ni(OH)₂ and the reconstructed substrate (Fig. S29). We have supplemented the corresponding description in the manuscript.

Fig. S25, Page 17; Fig. S29, Page 19

Fig. S25 XPS spectra of Mo 3d for Ni_{SA}-O/Mo₂C before and after HER and OER stability test.

Fig. S29 Ni 3d XPS spectra of Ni(OH)₂ and Ni_{SA}-O/Mo₂C after OER durability test.

Revision:

Left and right column, Page 7

“Fig. S25 shows the Mo 3d spectra of Ni_{SA}-O/Mo₂C after reactions, the major Mo peaks can be assigned to Mo⁶⁺.”

“However, for catalysts after OER, the Mo K-edge XANES spectra show more positive shifts compared with that of catalysts after HER, indicating catalysts after OER has higher valence state of Mo.”

Left column, Page 10

“In contrast, after OER (Fig. 5g), the position of the absorption edge of Ni K-edge spectra is higher than that of the Ni(OH)₂ reference, indicating the Ni has the higher

oxidation than +2, which probably due to the existence of NiOOH or the strong electron interaction between Ni(OH)₂ and the reconstructed substrate (Fig. S22).”

(4) What is the source of the surface oxide layer in the as-synthesized Mo₂C? Is this due to unintentional oxidation of Mo₂C during synthesis? How controlled is the oxidation level? The controllability and reproducibility of the oxide layer are important as nickel single atom is essentially linked with such oxygen. Further, why the intensity and area under the curves of Mo⁴⁺ and Mo⁶⁺ (related to surface MoO_x) are so high as compared to Mo²⁺ (related to Mo₂C) despite the sample being predominantly Mo₂C phase?

Response:

We highly appreciate the reviewer’s keen insight in this issue.

In fact, the degree of oxidation of Mo₂C can be controlled. H₂/Ar atmospheres has high reducibility. The low oxidation state of Mo²⁺ in the newly reduced Ni_{SA}-O/Mo₂C is easy to be oxidized. So, during we did experiment, we noticed the process of taking out the samples after pyrolysis can be adjusted to control the degree of oxidation of Mo₂C. To clarify the issue, we proposed two states of peroxidation and lower oxidation compared with our main sample of Ni_{SA}-O/Mo₂C, which was taken out from the tube furnace and exposed to the air at 30 °C. For the peroxidized sample, we take the newly reduced Ni_{SA}-O/Mo₂C out of the tube furnace and expose to the air at 60 °C. To get sample with lower oxidation, we saved the newly reduced Ni_{SA}-O/Mo₂C in Ar for 12 h. XPS is a surface inspection technology, which can be well used to see the oxidation state of the surface. Fig. S13a shows the comparison of the XPS spectra of Mo for all three samples. It can be observed that after the newly reduced Ni_{SA}-O/Mo₂C was treated with Ar environment for 12 h, the proportion of Mo²⁺ increased significantly, while the proportion of high oxidation state of Mo⁴⁺ and Mo⁶⁺ obviously decreased. In the peroxidized sample, the proportion of Mo²⁺ decreases obviously while the proportion of Mo⁴⁺ and Mo⁶⁺ increases significantly. We further calculated their area ratios of Mo²⁺ to high oxidation state Mo⁴⁺ and Mo⁶⁺ from XPS fitting results (Fig. S13b). The results show that the ratio is only 0.08 after peroxidation, and 0.42 after Ar environment through inert treatment, indicating that the Mo₂C surface has a strong oxidation behavior of Mo²⁺ when exposed to air at high temperature, while the Ar environment treatment prevented the oxidation behavior of Mo²⁺. In other words, the Mo₂C oxide layer comes from the oxidation of air, and different temperatures affect the intensity of the oxidation behavior. So, controlling the temperature at which the sample is taken from the tubular furnace and controlling the environment in which the sample is exposed in can control the level of surface oxidation of Mo₂C.

Indeed, just as the reviewer noticed the intensity and area under the curves of Mo⁴⁺ and Mo⁶⁺ (related to surface MoO_x) are higher as compared to Mo²⁺ (related to Mo₂C) despite the sample being predominantly Mo₂C phase. This is because the characterization method of XPS is a surface inspection technology, this just proves that the oxide is loaded on the surface of Mo₂C. To verify this, we performed XPS at different etching depths of 5 nm and 10 nm to explore the surface oxidation degree of the Ni_{SA}-O/Mo₂C samples (Fig. S13c). The XPS spectra of Mo 3d show that the

proportion of high oxidation state Mo^{4+} and Mo^{6+} decreases significantly with the increase of etching depth. At the etching depth of 5 nm, only a small amount of Mo^{6+} exists. However, there is basically only Mo^{2+} at the depth of 10 nm. At the same time, it can be observed from the XPS spectrum of C1s that Mo-C gradually increased with inward etching (Fig. S13d). All these results prove that the predominant phase of $\text{Ni}_{\text{SA}}\text{-O}/\text{Mo}_2\text{C}$ is Mo_2C , which surface is decorated by partially oxidized MoO_x .

Fig. S14, Page 12

Fig. S13 (a) High-resolution XPS spectra for Mo 3d. The $\text{Ni}_{\text{SA}}\text{-O}/\text{Mo}_2\text{C}$ was taken it out from tubular furnace after calcination at natural cooling to 30 °C. The peroxidation $\text{Ni}_{\text{SA}}\text{-O}/\text{Mo}_2\text{C}$ sample was taken it out at 60 °C. The $\text{Ni}_{\text{SA}}\text{-O}/\text{Mo}_2\text{C}$ -Ar was naturally cooled to room temperature and then taken it out after protecting it in Ar gas for 12 h; (b) The area ratio of $\text{Mo}^{2+}/(\text{Mo}^{4+}+\text{Mo}^{6+})$ from XPS; (c, d) High-resolution XPS spectra for Mo 3d and C 1s with an etching depth of 5 nm and 10 nm, respectively.

Revision:

Left column, Page 4

“It is worth mentioning that, the surface oxide layer can be controlled by the atmosphere and temperature to which the $\text{Ni}_{\text{SA}}\text{-O}/\text{Mo}_2\text{C}$ is exposed. For example, when the newly reduced $\text{Ni}_{\text{SA}}\text{-O}/\text{Mo}_2\text{C}$ were taken out of the tube furnace at 60 °C in air, its surface would be much more oxidized compared with the main sample of $\text{Ni}_{\text{SA}}\text{-O}/\text{Mo}_2\text{C}$. While, when the newly reduced $\text{Ni}_{\text{SA}}\text{-O}/\text{Mo}_2\text{C}$ were protected in Ar atmosphere for 12 h before taking out the furnace, its surface would be much lower oxidized compared with the main sample of $\text{Ni}_{\text{SA}}\text{-O}/\text{Mo}_2\text{C}$. (Fig. S13a-b). In addition, after etching the surface oxide layer, the main phase of Mo_2C gradually leaks out, which is demonstrated by XPS results with different etching depths (Fig. S13c-d).”

Revision in Supplement information:

Page 12-13

“The process of taking out the samples after pyrolysis can be adjusted to control the degree of oxidation of Mo₂C. The low oxidation state of Mo²⁺ in the newly reduced Ni_{SA}-O/Mo₂C is easy to be oxidized. Based on this, we proposed two states of peroxidation and lower oxidation. For the Ni_{SA}-O/Mo₂C, we take out from the tube furnace and exposed to the air at 30 °C. While we take the newly reduced Ni_{SA}-O/Mo₂C out of the tube furnace and expose to the air at 60 °C to get the peroxidized sample. To get sample with lower oxidation, we saved the newly reduced Ni_{SA}-O/Mo₂C in Ar for 12 h. As shown in Fig. S13a, after the newly reduced Ni_{SA}-O/Mo₂C was treated with Ar environment for 12 h, the proportion of Mo²⁺ increased significantly, while the proportion of high oxidation state of Mo⁴⁺ and Mo⁶⁺ obviously decreased. In the peroxidized sample, the proportion of Mo²⁺ decreases obviously while the proportion of Mo⁴⁺ and Mo⁶⁺ increases significantly. We further calculated their area ratios of Mo²⁺ to high oxidation state Mo⁴⁺ and Mo⁶⁺ from XPS fitting results (Fig. S13b). The results show that the ratio is only 0.08 after peroxidation, and 0.46 after Ar environment through inert treatment, indicating that the Mo₂C surface has a strong oxidation behavior when exposed to air at high temperature, while the Ar environment treatment prevented the oxidation behavior of Mo²⁺. In other words, the Mo₂C oxide layer comes from the oxidation of air, and different temperatures affect the intensity of the oxidation behavior. So, controlling the temperature at which the sample is taken from the tubular furnace and controlling the condition in which the sample is protected by an inert gas can control the level of surface oxidation of Mo₂C.

We further performed XPS at different etching depths of 5 nm and 10 nm to explore the surface oxidation degree of the Ni_{SA}-O/Mo₂C samples. Fig. S13a show that the proportion of high oxidation state Mo⁴⁺ and Mo⁶⁺ decreases significantly with the increase of etching depth. At the etching depth of 5 nm, only a small amount of Mo⁶⁺ exists. However, there is basically only Mo²⁺ at the depth of 10 nm. At the same time, it can be observed from the XPS spectrum of C1s that Mo-C gradually increased with inward etching (Fig. S13d). All these results prove that the predominant phase of Ni_{SA}-O/Mo₂C is Mo₂C, which surface is decorated by partially oxidized MoO_x.”

(5) The performance of HER and OER depends on both density of active sites and the intrinsic activity of those sites. Significant variation in ECSA indicates that there is a variation in the number of active sites among the samples. To clarify which sample has the best intrinsic activity, ECSA normalized LSV curves for both HER and OER need to be presented.

Response:

We are grateful for the reviewer’s valuable comments. Based on your suggestion, we first carefully read the relevant literatures and searching the appropriate method to determine their intrinsic electrocatalytic activity. ECSA normalized LSV curves for both HER and OER really can showed its intrinsic activity. Despite carbon materials have almost no electrocatalytic activity of their own, in electrocatalysis tests, carbon material is usually indispensable to drop-casted electrodes for ensuring smooth charge transfer at the catalyst interface and maximizing the utilization of the sample surface

(Fig. R1).^{r3} The purpose of getting the C_{dl} value purely contributed by $Ni_{ISA}\text{-O}/Mo_2C$, the electrode prepared by the drop-casting method does not include a conductive additive such as carbon black, because the high-surface-area carbon contributes to the non-faradaic current and interferes with the ECSA estimation. So, in order to ensure the authenticity of the data, we determined the C_{dl} data of a series of samples, including $Ni_{ISA}\text{-O}/Mo_2C$, $Ni_{1.5}\text{-O}/Mo_2C$, $Ni_{4.5}\text{-O}/Mo_2C$ and O/Mo_2C , in the same voltage ranges and under the same test conditions without adding carbon black.

As shown in Fig. S21, the $Ni_{ISA}\text{-O}/Mo_2C$ catalyst has the highest C_{dl} value and ECSA, suggesting that incorporation of Ni can significantly improve the density of active sites. As suggested, we have supplemented the ECSA normalized LSV curves for all electrocatalysts measured with the same test parameters. The specific activity (SA) normalized by ECSA in Fig. S22 shows that $Ni_{ISA}\text{-O}/Mo_2C$ exhibits the best electrocatalytic activity toward both HER and OER compared to other catalysts followed the order of $Ni_{ISA}\text{-O}/Mo_2C > Ni_{4.5}\text{-O}/Mo_2C > Ni_{1.5}\text{-O}/Mo_2C > O/Mo_2C$. In particular for OER, at the overpotential of 370 mV (Fig. S22d), $Ni_{ISA}\text{-O}/Mo_2C$ exhibits the SA of $8.71 \text{ mA cm}^{-2}_{ECSA}$, which is higher than those of $Ni_{4.5}\text{-O}/Mo_2C$ ($4.67 \text{ mA cm}^{-2}_{ECSA}$), $Ni_{1.5}\text{-O}/Mo_2C$ ($2.26 \text{ mA cm}^{-2}_{ECSA}$) and O/Mo_2C ($0.52 \text{ mA cm}^{-2}_{ECSA}$). These results indicate incorporation of Ni not only improves the density of active sites but also intrinsically improves HER and especially OER activity of $Ni_{ISA}\text{-O}/Mo_2C$ catalyst.

Fig. R1 Polarization curves of carbon black for HER and OER reaction.

Fig. S21-22; Page 15-16

Fig. S21 CVs were performed at various scan rates of 20, 40, 60, 80, and 100 mV s^{-1} : a) $\text{O}/\text{Mo}_2\text{C}$, b) $\text{Ni}_{1.5}\text{-O}/\text{Mo}_2\text{C}$; c) $\text{Ni}_{\text{SA}}\text{-O}/\text{Mo}_2\text{C}$, d) $\text{Ni}_{4.5}\text{-O}/\text{Mo}_2\text{C}$. e) Electrochemical active surface areas of $\text{O}/\text{Mo}_2\text{C}$ and $\text{Ni}_{\text{SA}}\text{-O}/\text{Mo}_2\text{C}$, $\text{Ni}_{1.5}\text{-O}/\text{Mo}_2\text{C}$, $\text{Ni}_{4.5}\text{-O}/\text{Mo}_2\text{C}$.

Fig. S22 (a-b) Polarization curves of $\text{Ni}_{\text{SA}}\text{-O}/\text{Mo}_2\text{C}$, $\text{Ni}_{1.5}\text{-O}/\text{Mo}_2\text{C}$, $\text{Ni}_{4.5}\text{-O}/\text{Mo}_2\text{C}$ and $\text{O}/\text{Mo}_2\text{C}$ for HER and OER normalized by ECSA. (c-d) Comparison of HER, OER specific activity of $\text{Ni}_{\text{SA}}\text{-O}/\text{Mo}_2\text{C}$, $\text{Ni}_{1.5}\text{-O}/\text{Mo}_2\text{C}$, $\text{Ni}_{4.5}\text{-O}/\text{Mo}_2\text{C}$ and $\text{O}/\text{Mo}_2\text{C}$ at 180 mV, 370 mV overpotential through polarization curves normalized by ECSA, respectively.

Revision:

Right column, Page 6 and Left column, Page 7

“Besides, the electrochemical active surface area (ECSA) of all samples was reflected from the double layer capacitance (C_{dl}) (Fig. S21). $\text{Ni}_{\text{SA}}\text{-O}/\text{Mo}_2\text{C}$ catalyst has the highest C_{dl} value, which indicates the $\text{Ni}_{\text{SA}}\text{-O}/\text{Mo}_2\text{C}$ has more exposed active surface area than other samples. Since the specific activity reflects the intrinsic activity of the catalysis, ECSA normalized LSV curves for both HER and OER were shown in Fig. S22. The specific activity (SA) normalized by ECSA shows that $\text{Ni}_{\text{SA}}\text{-O}/\text{Mo}_2\text{C}$ exhibits the best electrocatalytic activity toward both OER and HER compared to other catalysts followed the order of $\text{Ni}_{\text{SA}}\text{-O}/\text{Mo}_2\text{C} > \text{Ni}_{4.5}\text{-O}/\text{Mo}_2\text{C} > \text{Ni}_{1.5}\text{-O}/\text{Mo}_2\text{C} > \text{O}/\text{Mo}_2\text{C}$. In particular for OER, at the overpotential of 370 mV (Fig. S22 d), $\text{Ni}_{\text{SA}}\text{-O}/\text{Mo}_2\text{C}$ exhibits the SA of $8.71 \text{ mA cm}^{-2}_{\text{ECSA}}$, which is higher than those of $\text{Ni}_{4.5}\text{-O}/\text{Mo}_2\text{C}$ ($4.67 \text{ mA cm}^{-2}_{\text{ECSA}}$), $\text{Ni}_{1.5}\text{-O}/\text{Mo}_2\text{C}$ ($2.26 \text{ mA cm}^{-2}_{\text{ECSA}}$) and $\text{O}/\text{Mo}_2\text{C}$ ($0.52 \text{ mA cm}^{-2}_{\text{ECSA}}$). These results indicate incorporation of Ni not only improves the density of active sites but also intrinsically improves HER and especially OER activity of $\text{Ni}_{\text{SA}}\text{-O}/\text{Mo}_2\text{C}$ catalyst.”

References

r3. C. Wei, S. Sun, D. Mandler, X. Wang, S. Z. Qiao and Z. J. Xu, *Chem. Soc. Rev.*,

(6) Figure 5b shows that the Tafel slope of Ni_{4.5}/Mo₂C is the lowest (89.36 mV/Dec), however, in the text the same slope was assigned to Ni_{SA}-O/Mo₂C. Please check the same.

Response:

We greatly appreciate the reviewer for the careful reminder in these key figures, and we apologize for our carelessness and thoughtless. In fact, the Tafel slope graph belongs to Fig. 4b. Where the color and data of lines for Ni_{4.5}/Mo₂C and Ni_{SA}-O/Mo₂C are reversed. Now we have modified this Fig. 4b in revised manuscript.

Revision:

Fig. 4b, Page 6

Fig.4b Tafel plots for Ni_{SA}-O/Mo₂C, Ni_{1.5}-O/Mo₂C, Ni_{4.5}-O/Mo₂C and O/Mo₂C, commercial 20% Pt/C for OER.

(7) The oxidation of Mo₂C during OER is accountable as the catalytic surface can get oxidized via generated oxygen. But, what is the origin of oxidation of the catalysts during HER? It can be checked whether such oxidation happens spontaneously in alkaline conditions when the catalysts were dipped into the KOH electrolyte without doing any HER. This will clarify whether the oxidation of Mo₂C is automatic in alkaline conditions or is induced by the HER process.

Response:

We highly appreciate the reviewer for the enlightening comments. As suggested, we provided the XRD, Mo k-edge XANES and FT-EXAFS curve for Ni_{SA}-O/Mo₂C electrocatalysts after being dipped into the KOH electrolyte at open circuit potential (OCP). The X-ray diffraction (XRD) pattern of Ni_{SA}-O/Mo₂C electrocatalysts after being dipped into the KOH electrolyte at OCP showed the obvious characteristic peaks of Mo₂C phase (JCPDS card no.35-0787) and weak MoO₃ (JCPDS card no.05-0508 and 35-0609), obvious MoO₄²⁻ phase (JCPDS card no.29-1021), compared with the as-prepared Ni_{SA}-O/Mo₂C which only matched Mo₂C phase (Fig. S24). Based on the dissolution of MoO_x in KOH,⁴¹⁻⁴³ we speculate the original surface MoO_x reacted with

KOH to form MoO_4^{2-} , and with the consumption of surface MoO_x , Mo_2C surface is continuously oxidized to form new surface MoO_x until the Mo_2C is completely dissolved in KOH. The $\text{Ni}_{\text{ISA}}\text{-O}/\text{Mo}_2\text{C}$ after HER sample showed an XRD pattern similarity with that $\text{Ni}_{\text{ISA}}\text{-O}/\text{Mo}_2\text{C}$ electrocatalysts dipped into KOH electrolyte at OCP. The difference is that MoO_3 and MoO_4^{2-} phases are reduced. However, in the case of $\text{Ni}_{\text{ISA}}\text{-O}/\text{Mo}_2\text{C}$ after OER, all the XRD peaks could be assigned to the crystal structures of MoO_3 (JCPDS card no.05-0508 and 35-0609) and MoO_4^{2-} phase (JCPDS card no.29-1021) without Mo_2C peaks. These results are consistent with our HR-TEM observation (Fig. 5a, b), the $\text{Ni}_{\text{ISA}}\text{-O}/\text{Mo}_2\text{C}$ undergoes structural reorganization to form different phases after different electrochemical reactions. This phenomenon was further verified by carrying out XANES and FT-EXAFS measurement (Fig. 6c-e). Compared with $\text{Ni}_{\text{ISA}}\text{-O}/\text{Mo}_2\text{C}$ and $\text{Ni}_{\text{ISA}}\text{-O}/\text{Mo}_2\text{C}$ after HER, the absorption edge of $\text{Ni}_{\text{ISA}}\text{-O}/\text{Mo}_2\text{C}$ electrocatalysts dipped into KOH electrolyte at OCP was higher, suggesting that the average oxidation state of Mo was more positive. Meanwhile, its XANES shape is different from $\text{Ni}_{\text{ISA}}\text{-O}/\text{Mo}_2\text{C}$ after HER but similar with that of $\text{Ni}_{\text{ISA}}\text{-O}/\text{Mo}_2\text{C}$ after OER. Meanwhile, for the Mo k^3 -weighted FT-EXAFS spectra, its spectra display the existence of the Mo-O bond but no Mo-Mo bond like that of $\text{Ni}_{\text{ISA}}\text{-O}/\text{Mo}_2\text{C}$ after OER. These results indicate the dissolution and oxidation of Mo_2C in KOH is automatic in alkaline conditions. The applied oxidation potential of OER process will intensify these two continuous reaction processes, while HER process will inhibit to a certain extent the oxidation of the inner layer of Mo_2C and then partially prevent the formation of MoO_4^{2-} from the outer layer of MoO_x .

Revision:

Right column, Page 7 and Left column, Page 9

“In order to clarify whether the changes of $\text{Ni}_{\text{ISA}}\text{-O}/\text{Mo}_2\text{C}$ is automatic in KOH conditions, we tested the XRD, Mo k -edge XANES and FT-EXAFS curve for $\text{Ni}_{\text{ISA}}\text{-O}/\text{Mo}_2\text{C}$ electrocatalysts after being dipped into KOH electrolyte at open circuit potential (OCP). As shown in Fig. S24b, the $\text{Ni}_{\text{ISA}}\text{-O}/\text{Mo}_2\text{C}$ electrocatalysts after being dipped into KOH electrolyte at OCP ($\text{Ni}_{\text{ISA}}\text{-O}/\text{Mo}_2\text{C}$ -OCP) has similar characteristic peaks with $\text{Ni}_{\text{ISA}}\text{-O}/\text{Mo}_2\text{C}$ after HER except the MoO_3 and MoO_4^{2-} peaks are a little weaker. Based on the dissolution of MoO_x in KOH,⁴¹⁻⁴³ we speculate the original surface MoO_x reacts with KOH to form MoO_4^{2-} , and with the consumption of surface MoO_x , Mo_2C surface is continuously oxidized to form new surface MoO_x until the Mo_2C is completely dissolved in KOH. However, compared with $\text{Ni}_{\text{ISA}}\text{-O}/\text{Mo}_2\text{C}$ after OER, $\text{Ni}_{\text{ISA}}\text{-O}/\text{Mo}_2\text{C}$ -OCP has characteristic peaks of Mo_2C . In addition, compared with $\text{Ni}_{\text{ISA}}\text{-O}/\text{Mo}_2\text{C}$ and $\text{Ni}_{\text{ISA}}\text{-O}/\text{Mo}_2\text{C}$ after HER, the absorption edge of $\text{Ni}_{\text{ISA}}\text{-O}/\text{Mo}_2\text{C}$ -OCP was higher, suggesting that the average oxidation state of Mo was more positive. Meanwhile, the Mo k -edge XANES shape and the Mo k^3 -weighted FT-EXAFS spectra of $\text{Ni}_{\text{ISA}}\text{-O}/\text{Mo}_2\text{C}$ -OCP are different from $\text{Ni}_{\text{ISA}}\text{-O}/\text{Mo}_2\text{C}$ after HER but similar with that of $\text{Ni}_{\text{ISA}}\text{-O}/\text{Mo}_2\text{C}$ after OER. These results indicate the dissolution and oxidation of Mo_2C in KOH is automatic in alkaline conditions. The applied oxidation potential of OER process will intensify these two continuous reaction processes, while HER process will inhibit to a certain extent the oxidation of the inner layer of Mo_2C and then partially

prevent the formation of MoO_4^{2-} from the outer layer of MoO_x .”

Fig. S24, Page 17

Fig. S24 XRD characterization of $\text{Ni}_{\text{SA}}\text{-O}/\text{Mo}_2\text{C}$ catalyst (a) after HER and OER test; (b) as prepared/dipped into the KOH electrolyte at OCP.

Fig. 5c-e, Page 8

Fig 5 (c-e) *Ex situ* Mo k-edge XANES and the corresponding FT-EXAFS curve of $\text{Ni}_{\text{SA}}\text{-O}/\text{Mo}_2\text{C}$, $\text{Ni}_{\text{SA}}\text{-O}/\text{Mo}_2\text{C}$ -OCP and $\text{Ni}_{\text{SA}}\text{-O}/\text{Mo}_2\text{C}$ after HER/OER.

Added ref. 43-44

43. W.-F. Chen, C.-H. Wang, K. Sasaki, N. Marinkovic, W. Xu, J. T. Muckerman, Y. Zhu and R. R. Adzic, *Energy Environ. Sci.*, 2013, **6**, 943.

44. A. Aracena, A. Sanino and O. Jerez, *T NONFERR METAL SOC*, 2018, **28**, 177-185.

References

45. Z. He, J. Zhang, Z. Gong, H. Lei, D. Zhou, N. Zhang, W. Mai, S. Zhao and Y. Chen, *Nat. Commun.*, 2022, **13**, 2191.

(8) The surface of the catalysts after both HER and OER changes into MoO_x . However, it is well documented that molybdenum-based oxides are unstable in alkaline conditions [Trans. Nonferrous Met. Soc. China 28(2018) 177], and in fact, Molybdenum based chalcogenides are known to etch out through anodic oxidation process during OER process [Nature Communications 13 (2022) 2191]. It raises the question of whether the

catalyst possesses long-term stability as the data for only 15 hours is given. Long-term stability of at least 100 hours needs to be performed to check the stability of the catalyst and emphasize its importance in the field of water splitting.

Response:

Thank you for the insightful question. In addition, we also thank you for your suggestion of the two ref. (Trans. Nonferrous Met. Soc. China 28(2018) 177; Nature Communications 13 (2022) 2191). They are really useful for us. After carefully reading the two ref. and carefully thinking of your 7th comment, we have added some experiment data to better understand the process of structural change. So, just as you said, its electrochemical stability is indeed a problem worth thinking about. According to the reviewer's suggestion, we assembled the catalyst as the cathode and anode into a two-electrode electrolytic cell for overall water splitting stability test, respectively. As shown in inset of Fig. 4g, long-term electrolysis at a constant current has been performed under constant voltage of 1.69 V by chronoamperometry, the current density only decreased from 10 mA cm⁻² to 8.5 mA cm⁻² after 120 h, which indicates that the electrocatalyst can be used stably for a long time as a bifunctional electrocatalyst although the structure has been restructured during this process. We speculate this might be because, although the performance of HER process in half cell decreased slightly, the performance of OER process in half cell increased slightly. Their contributions cancelled each other out, so the overall stability was still good when used as a bifunctional catalyst.

Revision:

Left column, Page 7

“In the stability test, the current density has been only decreased from 10 mA cm⁻² to 8.5 mA cm⁻² after 120 h, indicating its good stability as a bifunctional electrocatalyst (inset in Fig. 4g).”

Inset in Fig. 4g, Page 6

Inset in Fig. 4g is the long-term durability of overall water splitting at a current

density of 10 mA cm^{-2} .

(9) The NiSA-O-Mo₂C is a pre-catalyst that undergoes oxidation, and the resultant Ni-Cluster-MoO_x is the main OER catalyst. Thus, in my view, the discussion of the high OER activity of NiSA-O-Mo₂C based on the d-band center is not suitable (page 8). The discussion of Ni-Cluster-MoO_x which was made in the later part may be sufficient and suitable.

Response:

Thank you for the reviewer's kind insight questions. Just as you said, NiSA-O/Mo₂C is a pre-catalyst, and the discussion in DFT simulations section about Ni_{cluster}/MoO_x is inadequate. As suggested, we supplemented the Gibbs free energy change of OER pathways, and the chemisorption energies of O₂ to discuss the OER activity of Ni_{cluster}/MoO_x. In addition, inspired by your question, we also calculated Gibbs free energy diagram of Volmer step for the reconstructed Ni_{cluster}/MoO_x during HER. These results were shown in Fig. 6h and Fig. S35. The supplemented and modified statements in page 11.

Fig. 6g, h, Page 9

Fig. 6 (g) Gibbs free energy diagram of Volmer step. Inset in (g): the optimal models of different stage of Ni_{SA}-MoO_x/Mo₂C. (h) Gibbs free energy change of OER pathways on Ni sites. Inset in (h): the atomic configurations of different stage of Ni_{cluster}/MoO_x.

Fig. S35, Page 21

Fig. S35 The chemisorption energies of O₂ of Ni_{SA}-O/Mo₂C and Ni_{cluster}/MoO_x.

Revision:

Right column, Page 10

“However, the O/Mo₂C substrate of Ni_{SA}-O/Mo₂C after HER and OER underwent oxidation and dissolved (Fig. 6g). For Ni_{SA}-MoO_x/Mo₂C after HER (Fig. 6e, S34), we can see that its energy barrier of water dissociation is 1.24 eV, which become larger than that of Ni_{SA}-O/Mo₂C, indicating the overoxidation of substrate can lead to some adverse hydrogen evolution effects. But for Ni_{cluster}/MoO_x after OER, the rate-determining step changes from the step pf *O to *OOH to the step of *OOH to O₂. Its free energy of the rate-determining step is 0.93 eV (Fig. 6e), which become smaller than that of Ni_{SA}-O/Mo₂C at U=1.23 V. In addition, compared with Ni_{SA}-O/Mo₂C, the reconstructed Ni_{cluster}/MoO_x shows weaker adsorption capacity of O₂ (Fig. S35), indicating the reconstructed Ni_{cluster}/MoO_x accelerates the oxygen evolution reaction. These results indicate the overoxidation of substrate is conducive to the aggregation of monatomic Ni into clusters and the improvement of oxygen evolution performance.”

(10) As per the EXAFS result, the coordination number of nickel (for Ni-O path) changes from 4 to 6 after both HER and OER, however, nickel is coordinated with less number of oxygen atoms in the DFT model ((Figure 6g). DFT should be done taking into consideration of same coordination environment as revealed by EXAFS.

Response:

We highly appreciate the reviewer’s careful review. Indeed, the coordination environment is very important for exploring the active site of single atom catalysts, we apologized for not think it through enough. Based on your questions and the comments from the reviewer #4’s, before modeling the new structures, we also added more oxygen atoms to simulate the top layer of oxides on Mo₂C even though we had introduced a small amount of oxygen atoms in the original models. As show in Fig. 6g, we have modified the inconsistency between the number of coordination atoms in models and the EXAFS results. And we modified the relevant description in the DFT simulations section.

Revision:

Fig. 6i, Page 9

Fig. 6i Structural reconstruction of Ni_{ISA}-O/Mo₂C catalyst after different reaction process. Water blue= Mo, green = Ni, grey = C, red = O.

(11) There are some general comments as well. The XANES and TEM were done before and after catalysis, however, in the abstract, it is mentioned that in-situ measurements were done, which is not true. In several sections the performance of the samples was overemphasized, for example, Ni_{ISA}-O-Mo₂C requires 1.64 V for an overall water-splitting current density of 10 mA/cm², which is only 0.05 V less as compared to Pt/C-IrO₂ couple, which should not be called ‘much better performance’, it is comparable only. Similarly, it has been mentioned in the abstract section that the HER performance of Ni_{ISA}-O-Mo₂C is ‘comparable to Pt/C’ which is not the case here, there is huge difference in their overpotentials as per Figure 4c. There are certain typo errors like ‘nationally designed’ in the abstract section, and ‘banding energy’ in the XPS section. So necessary correction should be done.

Response:

According to the reviewer’s important suggestion, we have revised the introduction and discussion part in order to make the manuscript have a more rigorous and accurate expression. We apologize again for our carelessness.

Revision:

Abstract, page 1, 3, 8, 11

“By *ex situ* synchrotron X-ray absorption spectroscopy and electron microscopy, we for the first time found, the configuration of atomically dispersed Ni in Ni_{ISA}-O/Mo₂C underwent reconstitution and O/Mo₂C support were oxidized with applied potentials.”

“By *ex situ* synchrotron X-ray absorption spectroscopy and electron microscopy observations, we for the first time found, the configuration of atomically dispersed Ni in Ni_{ISA}-O/Mo₂C electrocatalyst underwent transformation and O/Mo₂C support were overoxidized to varying degrees.”

“**Fig. 5** Structural analysis of the Ni_{ISA}-O/Mo₂C after HER/OER to study active sites. (a-b) HRTEM images of Ni_{ISA}-O/Mo₂C after HER/OER stability tests. (c-e) *Ex situ* Mo k-edge XANES of Ni_{ISA}-O/Mo₂C and the corresponding FT-EXAFS curve after HER/OER. (f-g) *Ex situ* XAS spectra of Ni K-edge of Ni_{ISA}-O/Mo₂C and the corresponding FT- EXAFS spectra after HER/OER, respectively. (h) HAADF-STEM image and EDS element mapping of Ni_{ISA}-O/Mo₂C after OER.”

“L. R. Z. guided the XANES, *ex situ* XAS measurements and analyzed XAS results.”

Left column, Page 7

“The result shows that the electrocatalytic overall splitting performance of Ni_{ISA}-O/Mo₂C||Ni_{ISA}-O/Mo₂C is comparable to that of Pt/C||IrO₂ especially under the high current density and superior to most reported single-atom catalysts (Table S6).”

Abstract, page 1

“In this work, non-noble-metal bifunctional catalysts with single Ni atom supported on oxygen-incorporated Mo₂C via Ni-O-Mo bridge bonds are rationally

designed and fabricated ($\text{Ni}_{\text{ISA}}\text{-O}/\text{Mo}_2\text{C}$),

Left column, page 4

“After loading of Ni atoms, the binding energy of Mo 3d XPS spectra for $\text{Ni}_{\text{ISA}}\text{-O}/\text{Mo}_2\text{C}$ display a positive shift about 0.12 eV compared with that of pure Mo_2C ,

Fig. 4, page 6

(a) Polarization curves and (b) Tafel plots for $\text{Ni}_{\text{ISA}}\text{-O}/\text{Mo}_2\text{C}$, $\text{Ni}_{1.5}\text{-O}/\text{Mo}_2\text{C}$, $\text{Ni}_{4.5}\text{-O}/\text{Mo}_2\text{C}$ and pure $\text{O}/\text{Mo}_2\text{C}$, IrO_2 for OER. (c) Polarization curve and (d) Tafel plots for $\text{Ni}_{\text{ISA}}\text{-O}/\text{Mo}_2\text{C}$, $\text{Ni}_{1.5}\text{-O}/\text{Mo}_2\text{C}$, $\text{Ni}_{4.5}\text{-O}/\text{Mo}_2\text{C}$ and pure $\text{O}/\text{Mo}_2\text{C}$, commercial 20% Pt/C for HER.

Response to Reviewer #2's Comments:

Chen and co-workers developed a high performance non-noble-metal bifunctional catalyst with single Ni atom supported on oxygen-incorporated Mo_2C via Ni-O-Mo bridge bonds. They for the first time found some interesting results. In different reactions, the atomic Ni underwent different microenvironment reconstitution. After HER, Ni species existed in atomically dispersed form. However, after OER, the atomically dispersed Ni were agglomerated into very small clusters by applied potential. So, the work is a nice discussion on the activity of single atom Ni on oxygen-incorporated Mo_2C starting from unprecedented synthesis, characterization electrocatalytic reactions and followed by the revelation of the post-reaction states of Ni. The discussion is valuable and easy to follow. Hence the manuscript is strongly proposed for acceptance in Nature Communications after considering the following minor revisions:

Response:

Thank you for the valuable professional comments that could help to further strengthen the manuscript. The concern raised by the reviewer and the valuable comments have been completely taken into account during the revision, and please see more details below.

1. The main sample is single Ni atom supported on oxygen-incorporated Mo_2C . The support Mo_2C were partially oxidized or oxygen-incorporated. How about the samples of oxygen-incorporated $\text{Ni}_{1.5}/\text{Mo}_2\text{C}$, $\text{Ni}_{4.5}/\text{Mo}_2\text{C}$ and pure Mo_2C ? Whether there is also partial oxidized in the surface of them? Please provide the data, such as Mo 3d XPS. In addition, if their surface were also oxygen-incorporated, I recommend the names of the three control samples should be change to $\text{Ni}_{1.5}\text{-O}/\text{Mo}_2\text{C}$, $\text{Ni}_{4.5}\text{-O}/\text{Mo}_2\text{C}$ and $\text{O}/\text{Mo}_2\text{C}$, respectively.

Response:

Thanks for the very valuable comments. Based on the author's comments, we tested the Mo 3d XPS of $\text{Ni}_{1.5}/\text{Mo}_2\text{C}$, $\text{Ni}_{4.5}/\text{Mo}_2\text{C}$ and pure Mo_2C . As shown in Fig. S15b, we can find the surface Mo_2C support in them were also partial oxidized to higher valence of +4 and +6. So, based on these data and reviewer's suggestion, we modified the original $\text{Ni}_{1.5}/\text{Mo}_2\text{C}$, $\text{Ni}_{4.5}/\text{Mo}_2\text{C}$ and pure Mo_2C to $\text{Ni}_{1.5}\text{-O}/\text{Mo}_2\text{C}$, $\text{Ni}_{4.5}\text{-O}/\text{Mo}_2\text{C}$

and O/Mo₂C. We have added corresponding change in relevant places in the manuscript.

Revision:

The whole manuscript, Page 3-6

For example, “As shown in Fig. 4a, the Ni_{SA}-O/Mo₂C exhibits an overpotential (η) of 299 mV at 10 mA cm⁻², which significantly exceeds those of Ni_{1.5}-O/Mo₂C (379 mV), Ni_{4.5}-O/Mo₂C (337 mV), pure O/Mo₂C (547 mV) and commercial IrO₂ catalyst (386 mV).” Left column in page 4.

Modified Fig. 2 and Fig. 4, page 4 and 6

Fig. 2 Structural characterization. (a) XPS spectra of Mo 3d. (b) XPS spectra of Ni 2p. (c) Mo K-edge XANES spectra. Inset in c) is the partial enlargement of the pre-edge peak. (d) The FT-EXAFS curves for the as-prepared samples. (e) Ni K-edge spectra. (f) The corresponding FT-EXAFS curve. (g) The wavelet transform (WT) of the Ni_{SA}-O/Mo₂C. (h-i) Ni K-edge EXAFS of Ni_{SA}-O/Mo₂C in R and k spaces.

Fig. 4 HER, OER and overall water splitting performance of the Ni_{SA}-O/Mo₂C catalyst in 1.0 M KOH. (a) Polarization curves and (b) Tafel plots for Ni_{SA}-O/Mo₂C, Ni_{1.5}-O/Mo₂C, Ni_{4.5}-O/Mo₂C and pure O/Mo₂C, commercial 20% Pt/C for OER. (c) Polarization curve and (d) Tafel plots for Ni_{SA}-O/Mo₂C, Ni_{1.5}-O/Mo₂C, Ni_{4.5}-O/Mo₂C and pure O/Mo₂C, IrO₂ for HER. (e) Comparison of the overpotentials of Ni_{SA}-O/Mo₂C and recently reported non-precious metal SACs at 10 mA cm⁻² in 1.0 M KOH. The histograms in yellow and green denote the HER, OER performance, respectively. (f) Current density versus time (*i*-*t*) curves for HER and OER. (g) Polarization curves of Ni_{SA}-O/Mo₂C||Ni_{SA}-O/Mo₂C and Pt/C||IrO₂ couples for overall water splitting in 1.0 M KOH. Inset (g) is the long-term durability of overall water splitting at a current density of 10 mA cm⁻². (h) Illustration of the Ni_{SA}-O/Mo₂C electrodes for overall water splitting.

Fig. S15b Mo 3d XPS spectra of O/Mo₂C, Ni_{1.5}-O/Mo₂C and Ni_{4.5}-O/Mo₂C, respectively.

2.How about the Ni 2p XPS spectra of Ni_{1.5}/Mo₂C catalyst?

Response:

Thanks for the comments. Based on the reviewer's comments, we have added Ni 2p XPS spectra of the Ni_{1.5}-O/Mo₂C sample, as show in Fig. S15a. Due to the Ni content in the Ni_{1.5}-O/Mo₂C is lower than Ni_{SA}-O/Mo₂C, the peak of Ni⁰ is not obvious, while Ni²⁺ is observed at 862.4 and 881.4 eV, indicating that Ni in the Ni_{1.5}-O/Mo₂C sample is almost entirely bonded to the surrounding oxygen atoms. We have added discourse in relevant places in the manuscript.

Revision:

Right column, Page 4

“It should be noted that, due to the Ni content in the Ni_{1.5}-O/Mo₂C is lower than Ni_{SA}-O/Mo₂C, the peak of Ni⁰ is not obvious, while Ni²⁺ is observed at 862.4 and 881.4 eV, indicating that Ni in the Ni_{1.5}-O/Mo₂C sample is almost entirely bonded to the surrounding oxygen atoms.”

Fig. S15, Page 13

Fig. S15a Ni 2p XPS spectra of Ni_{1.5}-O/Mo₂C and Ni_{4.5}-O/Mo₂C catalyst.

3. Figure S4 showed the N₂ adsorption-desorption isotherms, the specific areas increased after loading Ni in the surface, please explain this phenomenon.

Response:

Many thanks for the valuable question. Compared with O/Mo₂C, Ni_{SA}-O/Mo₂C was obtained by pyrolyzing the Mo-based organic nanosphere precursors with Ni. The Ni atom we introduced with nickel nitrate as Ni sources, and the nitrate ions would be decomposed and produce gas at high temperatures, which would carry away the carbon material in the nanospheres to a greater extent, resulting in more pores. So, we guess the increased specific surface areas probably come from the volatilization of nitrate from Ni sources. We have added the corresponding statement in revised manuscript.

Revision:

Right column, Page 3

“In addition, the specific surface areas obtained from N₂ adsorption-desorption isotherms show a significant increase from 14 for Mo-based precursor to 64 m² g⁻¹ for Ni_{SA}-O/Mo₂C (Fig. S8), which is probably due to the volatilization of nitrate from Ni sources.”

4. In Figure S1, the amorphous Mo-based precursor didn't look like smooth surface, the authors should provide accurate description.”

Response:

Thank for thoughtful and careful comments, and apologize for our inaccurate characterization. We have revised this manuscript very carefully and amend the unmatched descriptions.

Revision:

Left column, Page 3

“Transmission electron microscopy (TEM) images of Fig. S1 shows that the

amorphous Mo-based precursor is a nanosphere structure with rough surfaces with a diameter of about 200 nm.”

5. In Figure 5d-e, the disappearance of Mo-Mo diffraction peak after OER can be obviously observed. The author pointed out that it may be due to the further reaction between MoO₃ and KOH. Why did HER remain in MoO₃ state?

Response:

We highly appreciate the reviewer for the enlightening comments. Based on the Reviewer #1's comment 4, we carefully analyzed the XRD and XAFS spectra for Ni_{SA}-O/Mo₂C electrocatalysts after being dipped into the KOH electrolyte at open circuit potential (OCP) and after HER, OER test. The results showed the original surface MoO_x reacted with KOH to form MoO₄²⁻, and with the consumption of surface MoO_x, Mo₂C surface is continuously oxidized to form new surface MoO_x until the Mo₂C is completely dissolved in KOH. When no potential is applied, while the application of oxidation potential in OER process will accelerate the oxidation of the surface layer, resulting in rapid consumption of O/Mo₂C, which is the reason why the Mo₂C phase disappears after OER reaction. While, HER process will inhibit to a certain extent the oxidation of the inner layer of Mo₂C and then partially prevent the formation of MoO₄²⁻ from the outer layer of MoO_x, so the Ni_{SA}-O/Mo₂C after HER sample showed the diffraction peaks of Mo₂C, and little peak of MoO₄²⁻ and of MoO₃ (Fig. S24). The HR-TEM also confirms this view, the O/Mo₂C undergoes structural reorganization to form different phases due to instability in KOH and the influence of applied potential (Fig 5a, b). Through the XANES and FT - EXAFS measurement further verified that the sample after HER remain in MoO₃ state partly (Fig.5c-e). Meanwhile, the absorption edge of Ni_{SA}-O/Mo₂C electrocatalysts after HER significantly lower than that Ni_{SA}-O/Mo₂C after OCP and after OER, proving that the oxidation state is only slightly increased compared with Ni_{SA}-O/Mo₂C. More importantly, the Ni_{SA}-O/Mo₂C after HER display the existence of the Mo-Mo bond and not obvious Mo-O bond, further proving that the application of reduction potential in HER process prevents the further oxidation of Mo₂C and the reaction of MoO₃ with KOH.

Fig. S24, Page 17

Fig. S24 XRD characterization of Ni_{SA}-O/Mo₂C catalyst (a) after HER and OER test;

(b) as prepared/dipped into the KOH electrolyte at OCP.

Fig. 5c-e, Page 8

Fig 5 (c-e) *Ex situ* Mo k-edge XANES and the corresponding FT-EXAFS curve of Ni_{SA}-O/Mo₂C, Ni_{SA}-O/Mo₂C-OCP and Ni_{SA}-O/Mo₂C after HER/OER.

6. There are no pages in Ref. 15, 16, 20, 22 and 23, please check the whole ref. carefully.

Response:

We thank the reviewer for the comment and the very careful review. We have updated the page numbers of all the references to make them consistent in format.

Revision:

Page 11

3. I. Roger, M. A. Shipman and M. D. Symes, *Nat. Rev. Chem.*, 2017, **1**, 0003.
12. K. Jiang, M. Luo, M. Peng, Y. Yu, Y.-R. Lu, T.-S. Chan, P. Liu, F. M. F. De Groot and Y. Tan, *Nat. Commun.*, 2020, **11**, 2701.
15. Y. Zhu, W. Zhou, Y. Zhong, Y. Bu, X. Chen, Q. Zhong, M. Liu and Z. Shao, *Adv. Energy Mater.*, 2017, **7**, 1602122.
18. S. Yuan, M. Xia, Z. Liu, K. Wang, L. Xiang, G. Huang, J. Zhang and N. Li, *Chem. Eng. J.*, 2022, **430**, 132697.
22. M. Li, Y. Zhu, H. Wang, C. Wang, N. Pinna and X. Lu, *Adv. Energy Mater.*, 2019, **9**, 1803185.
24. B. H. R. Suryanto, Y. Wang, R. K. Hocking, W. Adamson and C. Zhao, *Nat. Commun.*, 2019, **10**, 5599.
25. F. Hu, D. Yu, M. Ye, H. Wang, Y. Hao, L. Wang, L. Li, X. Han and S. Peng, *Adv. Energy Mater.*, 2022, **12**, 2200067.
27. G. Li, H. Jang, S. Liu, Z. Li, M. G. Kim, Q. Qin, X. Liu and J. Cho, *Nat. Commun.*, 2022, **13**, 1270.
28. R. Li, P. Kuang, S. Wageh, A. A. Al-Ghamdi, H. Tang and J. Yu, *Chem. Eng. J.*, 2023, **453**, 139797.

7. Scale bar for the inset in Fig 6c should be shown.

Response:

We thank the reviewer for the comments and sorry for the missing some useful information data in figures. Based on the reviewer's comments, we have added scale bar in Fig. 6c.

In addition, inspired by this comment, we also checked the images for the whole article and added scale bar.

Revision:

Fig. 6c, Page 9

Fig. 6c PDOS graphs of 3d orbitals of Ni atoms in Ni_{SA}-O/Mo₂C and Ni_{SA}-Mo₂C. The black dash lines show the d-band center (ϵ), and the Fermi level was taken as zero of energy.

Fig. 1c-d, Page 2

Fig. 1 (c) HR-TEM image of Ni_{SA}-O/Mo₂C. (d) Elemental mapping of Mo, Ni, O, C for an individual Ni_{SA}-O/Mo₂C nanospheres (inset in Fig. 1c), respectively.

8. In Figure 4d, the significant numbers are inconsistent. In addition, “The cell are 1.69 V, 1.855 V and 1.9 V, while Pt/C||IrO₂ requires 1.64 V, 1.91V and 1.93V, respectively.

Response:

We thank the reviewer for the comment and the careful review. We apologize for our thoughtless and carelessness. Now we keep two significant digits in the data result and have modified it in the original manuscript.

Revision:

Right column, Page 3

“As show in Fig. 2a, the high-resolution Mo 3d spectra of pure Mo₂C were deconvoluted into six peaks, corresponding to Mo²⁺ (228.6 eV, 231.7 eV), Mo⁴⁺ (229.0 eV, 232.0 eV) and Mo⁶⁺ (232.8 eV, 235.8 eV) species.”

Left column, Page 7

“The cell voltage required to reach 10, 100 and 150 mA cm⁻² for Ni_{SA}-O/Mo₂C are 1.69 V, 1.86 V and 1.90 V, while Pt/C||IrO₂ requires 1.64 V, 1.91 V and 1.93 V, respectively.”

9. Recently some reported single atom catalysts for electrocatalysis should be cited, such as *Advanced Materials*, 2023, <https://doi.org/10.1002/adma.202303243>; *Adv. Funct. Mater.*, 2023, 2210867; *ACS Catalysis*, 2022, 12, 10771 – 10780.

Response:

We thank the reviewer for the comment and for providing the new relevant papers. The references provided by the reviewer have broadened our horizons on single atom catalysts. On the basis, we have cited the references in the manuscript for providing more inspirations for audience and further improving our manuscript.

Revision:

Added ref. 10, 16, 32

10. M. Wang, K. Sun, W. Mi, C. Feng, Z. Guan, Y. Liu and Y. Pan, *ACS Catalysis*, 2022, **12**, 10771-10780.

16. M. Li, H. Zhu, Q. Yuan, T. Li, M. Wang, P. Zhang, Y. Zhao, D. Qin, W. Guo, B. Liu, X. Yang, Y. Liu and Y. Pan, *Adv. Funct. Mater.*, 2022, **33**, 2210867.

32. P. Zhang, K. Chen, J. Li, M. Wang, M. Li, Y. Liu and Y. Pan, *Adv. Mater.*, 2023, **35**, 2303243.

Response to Reviewer #4's Comments:

In this work, the author designed and fabricated non-noble-metal bifunctional catalysts with single Ni atom supported on oxygen-incorporated Mo₂C via Ni-O-Mo bridge bonds. The HER and OER performance is analysed with DFT calculation and various experiments. However, it suffers from the follows problems and cannot be published in *Nature Communications*.

Response:

Thank you for the valuable yet professional comments that could help to further strengthen the manuscript. The concern raised by the reviewer and the valuable comments have been completely taken into account during the revision, and please see more details below.

(1) The electronic structure characterisation and calculation did not mention the spin state of Ni. It is well known that Ni has difference spin states and this is critical to understand its electronic structure, especially for oxides. Ni-O bonding cannot be well describe by PBE so I would suggest the author consider the spin polarized calculation with Hubbard model.

Response:

We highly appreciate the reviewer for the enlightening comments. And we are sorry that we omitted the difference spin states of Ni in the calculation. According to the reviewer's important suggestion, we performed the spin polarized calculation using

the Hubbard model for the $\text{Ni}_{\text{ISA}}\text{-O}/\text{Mo}_2\text{C}$ catalyst to explore the Ni-O bonding principle and the active source of the catalyst. At the same time, combined the comments from your comments 4, before calculating, we constructed new structures with more oxygen atoms added to simulate the top layer of oxides on Mo_2C even though we had introduced a small amount of oxygen atoms in the original models in our original manuscript. In addition, we modified the description of the DFT simulations section according to the new data. Please see modified figures, such as Fig. 3, Fig. 6, Fig. S19, Fig. 30 and Fig. 35 in revised manuscript and **Supplementary Information**.

Revision:

Supplementary Information in page 3:

Density Functional Theory Calculations.

“A series of density functional theory (DFT) calculations were all done with the Vienna Ab initio Simulation Package (VASP).^{1, 2} The spin polarized was considered and calculated with Hubbard model.”

Fig. 3, Page 5

Fig. 3 DFT calculation. The two plots of charge density differences of $\text{Ni}_{\text{ISA}}\text{-O}/\text{Mo}_2\text{C}$: top view of 3D plot (a) and 2D display (b). The isosurface level set to $0.003 \text{ e}\text{\AA}^{-3}$, where charge depletion and accumulation were depicted by purple and yellow, respectively. (c) PDOS graph of 3d orbitals of Ni, 4d orbitals of Mo and all orbitals of O atoms specified by cross marks in $\text{Ni}_{\text{ISA}}\text{-O}/\text{Mo}_2\text{C}$. (d) The chemisorption energies of Ni atom on surfaces of $\text{Ni}_{\text{ISA}}\text{-O}/\text{Mo}_2\text{C}$ and $\text{Ni}_{\text{ISA}}\text{-Mo}_2\text{C}$.

Fig. 6, Page 9

Fig. 6 DFT calculation of HER and OER. (a) The chemisorption energies of H^* and OH^* in Ni sites of $\text{Ni}_{\text{SA}}\text{-O}/\text{Mo}_2\text{C}$, $\text{Ni}_{\text{SA}}\text{-Mo}_2\text{C}$ and $\text{O}/\text{Mo}_2\text{C}$. Insets are their 3D plots of charge density differences: top view and side view. (b) The number of bader charges transfer at specified Ni, O and Mo atoms in $\text{Ni}_{\text{SA}}\text{-O}/\text{Mo}_2\text{C}$ and $\text{Ni}_{\text{SA}}\text{-Mo}_2\text{C}$. (c) PDOS graphs of 3d orbitals of Ni atoms in $\text{Ni}_{\text{SA}}\text{-O}/\text{Mo}_2\text{C}$ and $\text{Ni}_{\text{SA}}\text{-Mo}_2\text{C}$. The black dash lines show the d-band center (ϵ), and the Fermi level was taken as zero of energy. (d) The chemisorption energies of O in Ni sites of $\text{Ni}_{\text{SA}}\text{-O}/\text{Mo}_2\text{C}$ and $\text{Ni}_{\text{SA}}\text{-Mo}_2\text{C}$. (e,g) Gibbs free energy diagram of Volmer step. Inset in (g): the optimal models of different stage of $\text{Ni}_{\text{SA}}\text{-Mo}_x/\text{Mo}_2\text{C}$. (f,h) Gibbs free energy change of OER pathways on Ni sites. Inset in (h): the atomic configurations of different stage of $\text{Ni}_{\text{cluster}}/\text{MoO}_x$. (i) Structural reconstruction of $\text{Ni}_{\text{SA}}\text{-O}/\text{Mo}_2\text{C}$ catalyst after different reaction process. Water blue= Mo, green = Ni, grey = C, red = O.

Fig. S18 The plots of different optimal adsorption structures for oxygen on O/Mo₂C facet and several total energies (E_{DFT}).

Fig. S19, Page 14

Fig. S19 The plots of partially oxidized facet of Mo₂C (O/Mo₂C) and different optimal adsorption structures for nickel (Ni_{SA}-O/Mo₂C and Ni_{SA}-Mo₂C).

Fig. S30-35, Page 19-21

Fig. S30 The plots of charge density differences with Ni_{SA}-O/Mo₂C, Ni_{SA}-Mo₂C and O/Mo₂C, are, from top to down: top view of 3D plot and side view. The isosurface level set to 0.003 eÅ⁻³, where charge depletion and accumulation were depicted by purple and yellow, respectively.

Fig. S31 Gibbs free energy change of HER pathways on Ni sites of Ni_{SA}-O/Mo₂C, Ni_{SA}-Mo₂C and O/Mo₂C.

Fig. S32 Diagram of the optimal models for Volmer step at Ni_{SA}-O/Mo₂C and Ni_{SA}-Mo₂C, O/Mo₂C.

Fig. S33 Diagram of the optimal models for OER reaction step at Ni site of Ni_{SA}-O/Mo₂C.

Fig. S34 Diagram of the optimal models for Volmer step at Ni_{SA}-MoO_x/Mo₂C after HER-IT.

Fig. S35 The chemisorption energies of O₂ of Ni_{SA}-O/Mo₂C and Ni_{cluster}/MoO_x.

(2) The authors claim that "Ni–O–Mo bonding should dominate in the surface" (consistent with TEM image as in Figure 1), while in the atomic structure model only one Ni is placed in the supercell. The author did not provide the information about the supercell so I cannot estimate the Ni concentration but it is obvious far from that in the TEM image.

Response:

Thank you for the reviewer's interesting questions. We are sorry for the unclear expression about the active sites. In fact, single-atoms catalyst is a special supported catalyst that are featured with the individual active sites of SACs, which is composed of isolate metal atoms anchored by adjacent coordination species in the host materials. Meanwhile, in order to prevent the single atoms from agglomerating into particles, there is a strong interaction or considerable charge transfer between the individual metal atoms and the coordination species of the support, giving the individual metal atoms of SACs a unique electronic structure. So, the SACs have highly uniform active sites and geometric configuration, endowing them with similar electronic and spatial interaction with substrate, thereby achieving the enhanced catalytic selectivity. In simple terms, each active site on SACs is independent and homogeneous, so we only need to calculate one single active site during the DFT simulations section. So, even though, we can see all single metal atoms in TEM, in our simulation, we only need to build a model of a single atom according to the detail of experimental structure. This calculation pattern of calculating with a single active site is applicable to many carbon support as well as non-carbon support, including sulfide, hydroxide, phosphide, and oxide.^{r4-r11} To dispel your doubts, we provide the construction method of the supercell in the **Supplementary Information** for further understanding of the calculation process.

Reference:

r4. K. Qi, X. Cui, L. Gu, S. Yu, X. Fan, M. Luo, S. Xu, N. Li, L. Zheng, Q. Zhang, J. Ma, Y. Gong, F. Lv, K. Wang, H. Huang, W. Zhang, S. Guo, W. Zheng and P. Liu, *Nat. Commun.*, 2019, **10**. (Single-atom cobalt array bound to distorted 1T MoS₂ with ensemble effect for hydrogen

- evolution catalysis)
- r5. T. L. L. Doan, D. C. Nguyen, S. Prabhakaran, D. H. Kim, D. T. Tran, N. H. Kim and J. H. Lee, *AFM*, 2021, **31**, 2100233. (Single-Atom Co-Decorated MoS₂ Nanosheets Assembled on Metal Nitride Nanorod Arrays as an Efficient Bifunctional Electrocatalyst for pH-Universal Water Splitting)
- r6. Q. Wang, X. Huang, Z. L. Zhao, M. Wang, B. Xiang, J. Li, Z. Feng, H. Xu and M. Gu, *J. Am. Chem. Soc.*, 2020, **142**, 7425-7433. (Ultrahigh-Loading of Ir Single Atoms on NiO Matrix to Dramatically Enhance Oxygen Evolution Reaction)
- r7. H. Xu, T. Liu, S. Bai, L. Li, Y. Zhu, J. Wang, S. Yang, Y. Li, Q. Shao and X. Huang, *Nano Lett.*, 2020, **20**, 5482-5489. (Cation Exchange Strategy to Single-Atom Noble-Metal Doped CuO Nanowire Arrays with Ultralow Overpotential for H₂O Splitting)
- r8. Z. Lei, W. Cai, Y. Rao, K. Wang, Y. Jiang, Y. Liu, X. Jin, J. Li, Z. Lv, S. Jiao, W. Zhang, P. Yan, S. Zhang and R. Cao, *Nat. Commun.*, 2022, **13**, 24. (Coordination modulation of iridium single-atom catalyst maximizing water oxidation activity)
- r9. P. Li, M. Wang, X. Duan, L. Zheng, X. Cheng, Y. Zhang, Y. Kuang, Y. Li, Q. Ma, Z. Feng, W. Liu and X. Sun, *Nat Com*, 2019, **10**, 1711. (Boosting oxygen evolution of single-atomic ruthenium through electronic coupling with cobaltiron layered double hydroxides)
- r10. Q. Wang, Z. Zhang, C. Cai, M. Wang, Z. L. Zhao, M. Li, X. Huang, S. Han, H. Zhou, Z. Feng, L. Li, J. Li, H. Xu, J. S. Francisco and M. Gu, *J. Am. Chem. Soc.*, 2021, **143**, 13605-13615. (Single Iridium Atom Doped Ni₂P Catalyst for Optimal Oxygen Evolution)
- r11. K. Jiang, M. Luo, M. Peng, Y. Yu, Y.-R. Lu, T.-S. Chan, P. Liu, F. M. F. De Groot and Y. Tan, *Nat. Commun.*, 2020, **11**, 2701. (Dynamic active-site generation of atomic iridium stabilized on nanoporous metal phosphides for water oxidation)

Revision in Supplementary Information:

In page 5

Construction of structure:

Mo₂C: A 2x3 supercell surface consisting of 144 atoms from a cleaving surface of plane (1 0 1) of orthorhombic Mo₂C primitive cell was used firstly, and the fractional thickness set to 2.0, then adding 15 Å vacuum thickness to build vacuum slab crystal. The Brillouin zone integration is performed using the uniformly distributed scattering of going through the Gamma point to select a 2x2x1 k-mesh in the Monkhorst-Pack grid to make structure optimization, and we select 3x3x1 k-mesh to do density of state (DOS) calculations.

O/Mo₂C: A partially oxidized surface of Mo₂C is constructed by loading six oxygen atoms at the most stable adsorption site on the surface of Mo₂C.

Ni_{SA}-Mo₂C: A Ni atom coordinating with C atom is loaded in Mo₂C-O₆ surface.
Ni_{SA}-Mo₂C: The Mo₂C surface is loaded with more oxygen atoms to simulate a further oxidized surface, and then a nickel atom coordinating with four oxygen atoms is loaded on the surface.

Ni_{SA}-MoO_x/Mo₂C: The Mo₂C surface is loaded with more oxygen atoms to simulate a further oxidized surface and some of the carbon atoms are replaced by oxygen atoms to form a partially disordered MoO_x surface, then a nickel atom coordinating with six oxygen atoms is loaded on the surface.

$\text{Ni}_{\text{cluster}}/\text{MoO}_x$: The Mo_2C surface is loaded with more oxygen atoms to simulate a further oxidized surface and some of the carbon atoms are replaced by oxygen atoms to form a partially disordered MoO_x surface, then three nickel atoms coordinating with six oxygen atoms, respectively, are loaded on the surface.

To simulate the effect of inside a solid, the atoms located in the lower half of the slab were fixed for all calculation models.

(3) In Figure 1 d, the scale bar is missing. Is it the same particle as in Figure 1 c? Besides, the distribution of C is significantly different from other elements. The authors did not provide any explanation for this.

Response:

Thank you for the reviewer's perceptive question. We are sorry for the omissive scale bar, we have added the scale bar in Fig. 1d. Meanwhile, the element mapping image is from the same nanosphere as the inset in Fig. 1c, we have noted this detail in the legend. Based on the reviewer's comments, we examined the distribution of C in the sample, we can find that the C element are evenly dispersed compared with Mo. What's more, after Mo_2C is obviously oxidized by HER, OER reaction, C element is still uniformly distributed (Fig S27-28). C 1s spectra for the sample had not only the Mo-C bond, but also C=C and oxygenated functional groups, such as C-O, O-C=O (Fig. S14a). Meanwhile, XPS results showed that the content of C was more than four times that of Mo, the ratio is much higher than 2:1. Therefore, based on the above, we can reasonably speculate that Mo_2C nanocrystals grow on the carbon nanosphere. We carefully examined high-resolution TEM images of the sample, as shown in Fig. S10, where a carbon substrate was observed in the brighter regions of the sphere surface (highlighted by yellow circles), while Mo_2C nanocrystals with lattice streaks were present in the darker regions. In fact, it is common for Mo-containing organic precursors to generate Mo_2C by thermal decomposition^{R1,R2}. **We emphasized the presence of carbon sphere in the synthesis section.**

Revision:

Fig. 1c-d, Page 2

Fig. 1 (c) HR-TEM image of $\text{Ni}_{\text{SA-O}}/\text{Mo}_2\text{C}$. Insets in (b): SAED pattern. (d)

Elemental mapping of Mo, Ni, O, C for an individual Ni_{SA}-O/Mo₂C nanospheres inset in Fig. 1c, respectively.

Fig. S10, page 10

Fig. S10 HR-TEM images of the Ni_{SA}-O/Mo₂C.

Fig. S13, page 11

Fig. S14a XPS spectrum of the O-Mo₂C and Ni_{SA}-O/Mo₂C in the C 1s, respectively.

Table S2, page 22

Table S2 The content of each element in the Ni_{SA}-O/Mo₂C calculated by XPS.

Name	Mo	Ni	C	O
Atomic (%)	10.08	0.91	45.9	15

Revision:

Left column, Page 3

“During the pyrolysis process, the Mo-based organic precursors were simultaneously converted to Mo₂C nanocrystals with Ni ions fixed on their surfaces, which embedded in a uniform carbon substrate (Fig. S1).”

Right column, Page 3

“Energy-dispersive X-ray spectroscopy (EDS) mapping images reveal that Mo, Ni and O elements almost have the same double-shelled hollow shape and C elements is uniformly distributed in an individual Ni₅A-O/Mo₂C nanospheres (insert in Fig. 1c, and Fig. 1d). Combined with the HR-TEM magnified image in Fig. S10, we further confirm that Mo₂C nanocrystals with Ni ions were embedded in a uniform carbon substrate.”

Left column, Page 4

“The XPS spectrum of C1s (Fig. S14a) is fitted into four different signals at 283.9, 284.8, 285.8 and 288.9 eV, which are attributed to C-Mo, C-C/C=C, C-O, and O-C=O, respectively.”

The ref. r1-2

- r1. Z. Wu, J. Wang, R. Liu, K. Xia, C. Xuan, J. Guo, W. Lei and D. Wang, *Nano Energy*, 2017, **32**, 511-519.
- r2. S. Li, C. Cheng, A. Sagaltchik, P. Pachfule, C. Zhao and A. Thomas, *Adv. Funct. Mater.*, 2019, **29**, 1807419.

(4) The DFT simulation just considered the surface of MoC₂, without the top layer of oxides which is different from the TEM images. Is it a free surface as in DFT models or an interface as in TEM images?

Response:

Thank you for the insightful question. We are sorry for confusion caused. In fact, we took into account the oxide layer on the surface of Mo₂C in the DFT simulation, but in our original manuscript, we only considered the distribution of atoms around the Ni bonded part, which probably makes the concept of oxide layer not clear in modeling. Based on your suggestion and considering the surface of Mo₂C was partially oxidized, we referred to other literatures with similar structures (surface-oxidized CoP from J. Am. Chem. Soc. 2018, 140, 2610–2618; oxygen-Incorporated MoS₂ from J. Am. Chem. Soc. 2013, 135, 17881–17888 and o-doped CoP and o-doped CoP/CoO_x from J. Am. Chem. Soc. 2016, 138, 14686–14693) to added more O atoms to the surface of Mo₂C to simulate the amorphous oxide layer in order to get closer to partially surface oxidized Mo₂C with the existence of the MoO_x oxide layer in TEM results. Please see Fig. S19 of page 13 in supplemental information.

Revision:

Fig. S19, page 14

Fig. S19 The plots of partially oxidized facet of Mo₂C (O/Mo₂C) and different optimal adsorption structures for nickel (Ni_{SA}-O/Mo₂C and Ni_{SA}-Mo₂C).

(5) The authors conclude that OER leads to Ni cluster while HER needs Ni single atom as catalytic site, which means it cannot be bifunctional at the same time or alternatively.

Response:

Thank you for the insightful question, and we apologized for confusing the concept. The bifunctional electrocatalysts for HER and OER refers to pre-catalyst Ni_{SA}-O/Mo₂C without structural reconstruction in our manuscript. We can find when we assembled an electrolyzer with 1.0 M KOH electrolyte using Ni_{SA}-O/Mo₂C as bifunctional electrodes (Ni_{SA}-O/Mo₂C||Ni_{SA}-O/Mo₂C), the electrocatalytic overall splitting performance of Ni_{SA}-O/Mo₂C||Ni_{SA}-O/Mo₂C is comparable to that of Pt/C||IrO₂ especially under the high current density and superior to most reported single-atom catalysts (Table S6). Meanwhile, in the stability test, the current density has been only decreased from 10 mA cm⁻² to 8.5 mA cm⁻² after 120 h, indicating its good stability as a bifunctional electrocatalyst (inset in Fig. 4g) although the structure has been restructured during this process. We speculate this might be because, although the performance of HER process in half cell decreased slightly, the performance of OER in half cell process increased slightly. Their contributions cancelled each other out, so the overall stability was still good when used as a bifunctional catalyst. The surface reconstruction or irreversible structural are affected by many factors, such as applied potential, test environment. That is to say, during HER or OER tests, they each have different active sites to play a catalytic role. So, we think, the pre-catalyst Ni_{SA}-O/Mo₂C is a bifunctional catalyst, which can be used as both anode and cathode electrodes in a same electrolyzer.

(6) There are lots of typos, such as 'nationally designed' in the abstract.

Response:

We thank the reviewer for the comment and the careful review. We now have optimized the Abstract. In addition, we also checked the full text and corrected the inappropriate words.

Revision:

Abstract, page 1

“The rational design of efficient bifunctional single-atom electrocatalysts for

industrial water splitting and the comprehensive understanding of its complex catalytic mechanisms under harsh reaction conditions remain challenging. In this work, non-noble-metal bifunctional catalysts with Ni single atoms supported on oxygen-incorporated Mo₂C via Ni-O-Mo bridge bonds are rationally designed and fabricated (denoted as Ni_{SA}-O/Mo₂C), which gives high catalytic OER and HER performances. Remarkably, when used as a bifunctional electrocatalyst, Ni_{SA}-O/Mo₂C outperforms commercial Pt/C||IrO₂. By *ex situ* synchrotron X-ray absorption spectroscopy and electron microscopy, for the first time we found that the configuration of atomically dispersed Ni in Ni_{SA}-O/Mo₂C underwent reconstitution and O/Mo₂C support were oxidized and dissolved under applied potentials in KOH solution. After HER, the coordination number and bond lengths of Ni-O and Ni-Mo (Ni-O-Mo) were all altered, yet the Ni species still remain atomically dispersed. In contrast, after OER, besides the reconstitution of Ni-O and Ni-Mo (Ni-O-Mo), new Ni-Ni (Ni-O-Ni) bonds appeared, demonstrating the atomically dispersed Ni were agglomerated into very small clusters under applied potentials. Combining experimental results and DFT calculations, we infer that single-atom Ni on partially oxidized Mo₂C would be beneficial to HER; however, the small Ni clusters with optimal Ni-O, Ni-Mo and Ni-Ni coordination structures obtained after potential-driven reconstitution are more beneficial for OER. Therefore, the oxidation degree of Mo₂C and the configuration of single-atom Ni are both vital for HER or OER. This study provides both a feasible strategy and a new model to rational design highly efficient electrocatalysts for water electrolysis.”

Left column, page 4

“After loading of Ni atoms, the binding energy of Mo 3d XPS spectra for Ni_{SA}-O/Mo₂C display a positive shift about 0.12 eV compared with that of pure Mo₂C, ……”

(7) The Ni single atom catalyst's HER performance (the overpotential etc.) is not as good as Ni-doping in Mo₂C. Check Applied Catalysis B: Environmental 307, 121201, where Ni-doped Mo₂C gives a better performance.

Response:

We highly appreciate the reviewer’s valuable comment in this issue. We thank the reviewer for the comment. In fact, the performance of Ni_{SA}-O/Mo₂C catalyst is superior to most transition metal-based monatomic catalysts, however, to be honest, it still has room for improvement when compared of some self-supporting structures with nickel foam and carbon paper. This kind of self-supporting structure is difficult to rule out whether the substrate has produced additional help to the catalyst, and even its support structure will provide some performance. This comparison with the performance of powdered electrocatalysts is not quite fair. In this revised **Supplemental Information**, we have added more recently published single-atom catalysts to demonstrate the comparable catalytic HER performance of Ni_{SA}-O/Mo₂C (Table S4,5,6). It worth noting that, in our manuscript, even though its performance is not the best compared with the reported catalysts, we pay much more attention to explore the structural transformation of single Ni catalytic sites during catalysis, and further explore the structure-activity relationship between structure and catalytic performance. We think our study can provides both a feasible strategy and a new model to rational design

highly efficient electrocatalysts for water electrolysis. Thanks for your support and patience.

Revision:

Supplemental Information

Table S4-6, page 24-25

Table S4 Comparison of OER performance of Ni_{SA}-O/Mo₂C and other reported non-precious metal single-atom electrocatalysts.

Electrocatalysts	Electrolyte	η_{10} (mV vs. RHE)	Tafel slope (mV dec ⁻¹)	Working electrode	Ref
Ni _{SA} -O/Mo ₂ C	1.0 M KOH	299	89	carbon paper	This work
Co1/TaS ₂	1.0 M KOH	330	70	glassy-carbon	ACS Nano. 2021, 15, 7105–7113.
NiPc-GO	1.0 M KOH	320	61	glassy-carbon	ACS Nano. 2020, 14, 13279-13293.
HCM@Ni-N	1.0 M KOH	304	76	carbon paper	Adv. Mater. 2019, 31, 1904548.
CoNi-SAs/NC	1.0 M KOH	340	58.7	carbon cloth	Adv. Mater. 2019, 31, 1905622.
NiSO ₄ -GF	1.0 M KOH	300	80	graphite foil	ACS Nano. 2020, 14, 11662-11669.
NiFe-CNG	1.0 M KOH	270	74	glassy-carbon	Nat. Commun. 2021, 12, 1-13.
Co SAs/Mo ₂ C	1.0 M KOH	270	74.9	carbon cloth	J. Mater. Chem. A, 2020, 8, 3071–3082.
Ni-NHGF	1.0 M KOH	331	63	glassy-carbon	Nat. Catal. 2018, 1, 63- 72.
Co-NHGF	1.0 M KOH	402	80	glassy-carbon	Nat. Catal. 2018, 1, 63- 72.
Fe-NHGF	1.0 M KOH	488	175	glassy-carbon	Nat. Catal. 2018, 1, 63- 72.
Co-POC	0.1 M KOH	470	139	glassy-carbon	Adv. Mater. 2019, 31, 1900592.

Fe/N-G-SAC	0.1 KOH	M	370	73	glassy-carbon	Adv. Mater. 2020, 32, 2004900.
Fe-N ₄ SAs/NPC	1.0 KOH	M	430	95	glassy-carbon	Angew. Chem., Int. Ed., 2018, 57, 8614–8618
CoSA/N,S-HCS	1.0 KOH	M	306	38	glassy-carbon	Adv. Energy Mater. 2020, 10, 2002896
Ni SAs@S/N-CMF	1.0 KOH	M	285	50.8	carbon paper	Adv. Mater. 2203442.
Ni-N,P/CNFs	1.0 KOH	M	330	65	carbon cloth	Nano Energy, 2022, 98, 107266.
Ni-N,S/CNFs	1.0 KOH	M	350	111	carbon cloth	Nano Energy, 2022, 98: 107266.
CoSAs-MoS ₂ /TiN NRs	1.0 KOH	M	340.6	81.2	carbon cloth	Adv. Funct. Mater., 2021, 31, 2100233.

Table S5 Comparison of HER performance for Ni_{SA}-O/Mo₂C single-atom catalysts and non-precious metal single-atom catalysts reported in the literature.

Electrocatalysts	Electrolyte	η_{10} (mV vs. RHE)	Tafel slope (mV dec ⁻¹)	Working electrode	Ref
Ni _{SA} -O/Mo ₂ C	1.0 M KOH	133	83.6	glassy-carbon	This work
CoSA/N,S-HCS	1.0 M KOH	165	96	glassy-carbon	Adv. Energy Mater. 2020 , 10, 2002896
Mo ₁ N ₁ C ₂	0.1 M KOH	132	90	glassy-carbon	Angew. Chem., Int. Ed., 2017 , 56, 16086–16090
Fe-N ₄ SAs/NPC	1.0 M KOH	202	123	glassy-carbon	Angew. Chem., Int. Ed., 2018 , 57, 8614–8618
Co SAs/Mo ₂ C	1.0 M KOH	178	155	carbon cloth	J. Mater. Chem. A, 2020 , 8, 3071–3082
Co-BM-C	1.0 M KOH	126	81	Ni foam	Chem. Eng. J. 2021 , 433, 134089.
Mo-Co ₉ S ₈ @C	1.0 M KOH	113	67.6	carbon paper	Adv. Energy Mater. 2020 , 10, 1903137.

CoSAs-MoS ₂ /TiN NRs	1.0 M KOH	131.9	56.9	carbon cloth	Adv Funct Mater, 2021 , 31,2100233.
NiCo DASs/N-C	1.0 M KOH	189	72.5	glassy-carbon	Adv. Funct. Mater. 2022 , 2210867
CoNC-SA/N*-C	1.0 M KOH	194	91.9	glassy-carbon	ACS Catal. 2022 , 12, 10771–10780
SC-CuSA-NC	1.0 M KOH	124	107.5	glassy-carbon	Composites Part B 2023 , 253, 110575
Co SAs-Co NPs/NCFs	1.0 M KOH	205	83.2	glassy-carbon	Journal of Energy Chemistry 2022 , 67, 147-156
120Ni-MSAC	1.0 M KOH	190	83.5	glassy-carbon	Chemical Engineering Journal 2023 , 468, 43733
CoNC-SA/N*-C	1.0 M KOH	194	91.9	glassy-carbon	ACS Catal. 2022 , 12, 10771–10780

Table S6 Comparison of water splitting performance for Ni_{SA}-O/Mo₂C and others single-atom catalysts reported in the literature at current densities of 50, 100, 100 mA cm⁻¹.

Electrocatalysts	Electrolyte		50 (mA cm ⁻¹)	100 (mA cm ⁻¹)	200 (mA cm ⁻¹)	Working electrode	Ref
Ni _{SA} -O/Mo ₂ C Ni _{SA} -O/Mo ₂ C	1.0 KOH	M	~1.79	1.85V	1.93	carbon cloth	This work
Ir ₁ @Co/NC Ir ₁ @Co/NC	1.0 KOH	M	~1.85			carbon paper	Angew. Chem. Int. Ed., 2019, 58,2-8.
Fe-N ₄ SAs/NPC Fe-N ₄ SAs/NPC	1.0 KOH	M	~2.0			carbon paper	Angew. Chem., Int. Ed., 2018, 57, 8614–8618
CoSAs- MoS ₂ /TiN NRs	1.0 KOH	M	2.02			carbon cloth	Adv Funct Mater., 2021, 31,2100233.
Co-BM-C Co- BM-C	1.0 KOH	M	~1.88	~1.91	~2.1	Ni foam	Chem. Eng. J., 2021,134089.
Rh SAC-CuO NAs/CF Rh SAC-CuO NAs/CF	1.0 KOH	M	~1.65	~1.7	1.77	copper foam	Nano letters, 2020, 20(7): 5482- 5489.

(8) The final state in Figure 6 (e) is missing. Why are they different from each other?

Response:

We thank the reviewer for the careful review. We apologize for our carelessness and thoughtless. Based on the reviewer's comments, we have added the reaction intermediates states in Fig. 6e. In the DFT calculation of the water dissociation step, the difference of construction models of each sample lead to different final state energy, which is obviously different from free energy of H* adsorption although their process is also divided into three stages. We guess the reviewer mistook the diagram for the free-energy diagrams for the HER like in Fig. S31, in which they have the same final state energy. If our answer is not what you want, please let us know. Thanks again for your patience.

Revision:

Fig. 6e, Page 9

Fig. 6e Gibbs free energy diagram of Volmer step.

Fig. S31 Gibbs free energy change of HER pathways on Ni sites of Ni_{SA}-O/Mo₂C, Ni_{SA}-Mo₂C and O/Mo₂C.

In summary, we have carefully modified the entire manuscript. All the changes have been marked with **yellow highlight** in the revised manuscript and Supporting Information. Now we submit the revised version to *Nature Communication*. We look forward to your reply. Thank you very much.

Sincerely yours

Chen Chen
Department of Chemistry
Tsinghua University

Reviewers' Comments:

Reviewer #1:

Remarks to the Author:

The authors have addressed most of the concerns by incorporating suggestions and clarifications with additional results. In my view, the manuscript can be accepted for publication after addressing some minor comments as given below

(1) As per my earlier comments, authors have now clarified that the synthesized material is Ni-SA-O-Mo₂C embedded within carbon matrix. Since, nickel precursor was loaded to the Mo-containing organic precursor, it is very much unlikely that Ni-SA selective link with only in-situ generated Mo₂C. There is also a significant possibility of Ni-SA linking with carbon matrix, which can show its own electrocatalytic activity. Author should include this possibility in the discussion of electrocatalytic activity.

(2) The discussions of XPS spectra after HER and OER are still not adequate. The shift of Mo₆₊ peak (Figure S25) towards higher energy after OER is accountable due to over oxidation, however, it is not clear why the peak shifts to lower energy after HER (as compared to original sample), despite its oxidation. As oxidation happens in both HER and OER, thus Mo₆₊ peak should shift to higher energies in both the cases (as compared to original sample), with more shift for the OER sample. This is precisely the trend as observed from EXAFS.

(3) Figure S13c (XPS depth profile) is the most important to prove that the catalyst contains MoO_x at the surface and Mo₂C at the core, which could confirm the claim that Ni-SA linked with the surface oxide layer. Thus, I suggest to move this very informative figure to the main manuscript.

Reviewer #2:

Remarks to the Author:

The authors have revised and answered all the questions. Accept is OK!

Reviewer #4:

Remarks to the Author:

The authors have answered all the questions and the paper is ready to be published as it is.

Response to Reviewers' Comments and Revisions Made

Response to Reviewer #1's Comments:

The authors have addressed most of the concerns by incorporating suggestions and clarifications with additional results. In my view, the manuscript can be accepted for publication after addressing some minor comments as given below

Response:

We appreciate the reviewer's positive as well as the valuable yet professional comments that could help to further strengthen the manuscript. The concern raised by the reviewer and the valuable comments have been completely taken into account during the revision, and please see more details below

(1) As per my earlier comments, authors have now clarified that the synthesized material is Ni-SA-O-Mo₂C embedded within carbon matrix. Since, nickel precursor was loaded to the Mo-containing organic precursor, it is very much unlikely that Ni-SA selective link with only in-situ generated Mo₂C. There is also a significant possibility of Ni-SA linking with carbon matrix, which can show its own electrocatalytic activity. Author should include this possibility in the discussion of electrocatalytic activity.

Response:

We highly appreciate the reviewer's valuable comment in this issue. We completely agree that the current experimental data cannot rule out whether carbon matrix contributes to the electrocatalytic activity. So, in order to clarify the electrocatalytic activity of Ni-SA linking with carbon matrix, we prepared two control samples. The first, Ni_{SA}-O/Mo₂C was ultrasonically dissolved in 8 M HNO₃ under constant temperature water bath at 50 °C for 36 h with continuous stirring, and then washed to neutral to remove in-situ generated Mo₂C particles (denoted Ni_{SA}/C_{etch}). The second, pure nanospheres were prepared according to the same synthesis method with Ni_{SA}-O/Mo₂C except using only C₆H₁₂O₆ and CH₄N₂S, and then Ni metal ions were adsorbed on their surface and finally calcined them at high temperature (denoted as Ni/C). It can be seen from XRD patterns that both Ni_{SA}/C_{etch} and Ni/C show carbon peaks, that is to say, control samples have no Mo₂C nanoparticles in the sample (Fig. S20a,b). Meanwhile, TEM and element mapping can also confirm the absent of Mo₂C nanoparticles and the uniform distribution of Ni elements on carbon matrix (Fig. S20c, d). Subsequently, we performed electrochemical tests on Ni_{SA}/C_{etch} and Ni@C under the same test conditions and catalyst loading. It can be seen from the LSV curves that neither Ni_{SA}/C_{etch} nor Ni/C show good catalytic activity, that is to say, the catalytic activity of Ni_{SA}-O/Mo₂C comes from the Ni single atoms anchored on oxygen-incorporated Mo₂C (Fig. S21).

Revision:

Right column, Page 6

It is worth noting that in order to exclude the role of Ni atoms on carbon substrate, we set up two control samples: Ni atoms on carbon substrate obtained via etching Mo₂C particles in Ni_{SA}-O/Mo₂C (Ni_{SA}/C_{etch}) and Ni atoms on pure carbon spheres (Ni/C) synthesized by the same method with Ni_{SA}-O/Mo₂C. As can be seen from XRD patterns, TEM and element mappings (Fig. S20), both samples only show carbon peaks and uniform Ni element distribution. Electrochemical tests showed that, compared with Ni_{SA}-O/Mo₂C, their HER and OER catalytic activity can be ignored (Fig. S21), indicating the catalytic activity of Ni_{SA}-O/Mo₂C comes from the Ni single atoms anchored on oxygen-incorporated Mo₂C.

Fig. S20, page 15

Fig. S20 (a, b) XRD patterns of Ni_{SA}/C_{etch} and Ni/C; (c, d) HR-TEM image of Ni_{SA}/C_{etch} and Ni/C, respectively. Insets in (c): SAED pattern, and elemental mapping of Ni, O, C and Mo for an individual nanospheres.

Fig. S21, page 15

Fig. S21 (a-b) Polarization curves of Ni/C and Ni_{SA}/C_{etch} for HER and OER.

(2) The discussions of XPS spectra after HER and OER are still not adequate. The shift of Mo⁶⁺ peak (Figure S25) towards higher energy after OER is accountable due to over oxidation, however, it is not clear why the peak shifts to lower energy after HER (as compared to original sample), despite its oxidation. As oxidation happens in both HER and OER, thus Mo⁶⁺ peak should shift to higher energies in both the cases (as compared to original sample), with more shift for the OER sample. This is precisely the trend as observed from EXAFS.

Response:

We highly appreciate the reviewer's valuable comment in this issue. We are sorry that this problem may not have been described clearly before. In Mo XPS spectra after HER and OER, we can find, the major Mo peaks can be assigned to Mo⁶⁺ without low valence state of Mo, suggesting the average oxidation state of Mo are both higher than that of Ni_{SA}-O/Mo₂C, that is to say the surface of the catalysts both have obvious oxidation behavior. However, after HER reaction, the Mo⁶⁺ peak shifted to a little lower position, which is probably due to the interaction between single atom Ni and the reconstructed substrate. Therefore, just as the reviewer said, as precisely shown in Mo K-edge XANES spectra, after HER and OER, the average oxidation states of Mo both increased. And, for catalysts after OER, the Mo K-edge XANES spectra show more positive shifts compared with that of catalysts after HER, indicating catalysts after OER has higher average valence state of Mo, which consistent with XPS results. We have added the corresponding description in the revised manuscript.

Revision:

Right column, Page 7

Fig. S27 shows the Mo 3d spectra of Ni_{SA}-O/Mo₂C after reactions, the major Mo peaks can be assigned to Mo⁶⁺ without low valence state of Mo, suggesting the average oxidation state of Mo are both higher than that of Ni_{SA}-O/Mo₂C, that is to say the surface of the catalysts both have obvious oxidation behavior. However, after HER reaction, the Mo⁶⁺ peak shifted to a little lower position, which is probably due to the interaction between single atom Ni and the reconstructed substrate.

(3) Figure S13c (XPS depth profile) is the most important to prove that the catalyst contains MoO_x at the surface and Mo₂C at the core, which could confirm the claim that Ni-SA linked with the surface oxide layer. Thus, I suggest to move this very informative figure to the main manuscript.

Response:

We thank the reviewer for the good suggestion. Based on your suggestion, we have moved original Figure S13c and merged all the data into Fig. 2a in revised manuscript, which really can more intuitively see the existence of oxidation layer on the surface of Mo₂C and further explain that Ni-SA linked with the surface oxide layer.

Revision:

Fig. 2a, page 4

Fig. 2 Structural characterization. (a) XPS spectra of Mo 3d.

Reviewer #2 (Remarks to the Author):

The authors have revised and answered all the questions. Accept is OK!

Response:

Thanks for the positive comments.

Reviewer #4 (Remarks to the Author):

The authors have answered all the questions and the paper is ready to be published as it is.

Response:

Thanks for the positive comments.